# Extraction of Pre-earthquake Anomalies in Borehole Strain Data Using Graph WaveNet: A Case Study of the 2013 Lushan Earthquake, China

Chenyang Li [1, 4], Yu Duan[1, 4], Ying Han[2], Zining Yu[3], Chengquan Chi[1, 4, *] and Dewang Zhang[1, 4, *]

[1] School of Information Science and Technology, Hainan Normal University, Haikou, 571158, China
[2] College of Geography and Environmental Science, Hainan Normal University, Haikou, 571158, China
[3] Ocean University of China, College of information science and engineering, Qingdao, 266100, China
[4] Key Laboratory of Data Science and Smart Education, Hainan Normal University, Ministry of Education

*Correspondence to*: Chengquan Chi (chicqhainnu@gmail.com), Dewang Zhang (zhangdwsan@163.com);

**Abstract.** On 20 April 2013, Lushan experienced a magnitude 7.0 earthquake. In seismic assessments, borehole strain meters, recognized for their remarkable sensitivity and inherent reliability in tracking crustal deformation, are extensively employed. However, traditional data processing methods encounter challenges when handling massive datasets. This study proposes using a graph wavenet graph neural network to analyze borehole strain data from multiple stations near the earthquake epicenter and establishes a node graph structure using data from four stations near the Lushan epicenter, covering

years 2010–2013. After excluding the potential effects of pressure, temperature, and rainfall, we statistically analyzed the pre-earthquake anomalies. Focusing on the Guza, Xiaomiao, and Luzhou stations, which are the closest to the epicenter, the fitting results revealed two accelerations of anomalous accumulation before the earthquake. Approximately four months before the earthquake event, one acceleration suggests the pre-release of energy from a weak fault section. Conversely, the acceleration a few days before the earthquake indicated a strong fault section reaching an unstable state with accumulating

strain. We tentatively infer that these two anomalous cumulative accelerations may be related to the preparation phase for a large earthquake. This study highlights the considerable potential of graph neural networks in conducting multi-station studies of pre-earthquake anomalies.

## 1 Introduction

Earthquakes result from the accumulation of stress in the Earth's crust during plate movement and collisions. Once the stress

surpasses a critical threshold, the crust ruptures, unleashing seismic waves that reverberate through the ground, causing substantial damage (Campbell et al., 2020; Fan et al., 2021). Extensive research on earthquakes has generated a wealth of information worldwide, establishing a robust database for studying pre-earthquake anomalies.

On 20 April 2013, 08:02 UTC, a magnitude 7.0 earthquake struck Lushan County, Ya'an, Sichuan, China, within the Longmenshan fault zone. The epicenter was located at approximately 103° E and 30.30° N within the Longmenshan Fault

Zone at a depth of 13 km. The earthquake mechanism solution revealed a retrograde rupture of the earthquake-induced

rupture. By 24 April 2013, the earthquake had triggered over 4,000 aftershocks, affecting more than 300,000 people across an expansive area exceeding 12,000 km$^2$. The aftermath witnessed a significant loss of life and property. Additionally, the seismic event triggered various geological hazards, such as earth fissures, landslides, and surface deformation (Hong et al., 2013).

Researchers around the world have examined various phenomena preceding and following earthquakes, delving into subterranean, surface, and spatial changes. Chen et al., (2014) studied the co-seismic ionospheric anomalies of the Lushan earthquake. Guo and Zheng, (2022) calculated and analyzed the anomalies of background noise near the pre-earthquake epicenter of Lushan earthquake. Liu et al., (2014b) analyzed the aerosol optical depth (AOD) and concluded that the AOD could be a potential earthquake precursor in the Sichuan Basin. Liu et al. (2014a) examined groundwater anomalies and

identified medium-term dynamics and short-term or impending anomalies in solid tidal aberrations of the water-level. Ma et al., (2015) analyzed pre-earthquake tidal cycles and concluded that celestial tidal forces trigger earthquakes under critical rock fragmentation and sliding conditions. Zhang et al., (2016) explored thermal anomalies as a precursor to earthquakes using a time series of surface temperatures prior to the Lushan earthquake. Zhu et al., (2013) analyzed the tectonic deformation and energy accumulation in the southern section of the Longmenshan Fracture Zone through mobile gravity

observation data. Jiang et al., (2013) found severe negative anomalies before and on the day of the earthquake by analyzing the vertical total electron content (VTEC) anomaly in the ionosphere.

    Based on the findings from the United States Plate Boundary Observation (PBO) project proposal, borehole strain observations have emerged as superior to GPS and laser strain meters in capturing short- to medium-term, as well as pre-earthquake, strain variations (Zhang, 2004; Zheng and Zhang, 2004). China has deployed multiple YRY-4 four-component

borehole strainmeters, offering not only four-component data but also auxiliary observations of air pressure, groundwater level, and temperature (Chi et al., 2007; Qiu, 2014; Qiu et al., 2020). Numerous studies on the Lushan earthquake have employed borehole strain data. Qiu et al., (2013) correlated borehole strain data from the Guza station before the Lushan Ms7.0 earthquake with other influencing factors, establishing a connection to the earthquake. Zhu et al., (2018) identified the precursors of the Lushan earthquake by analyzing the eigenvalues and eigenvectors from borehole data. Yu et al., (2021)

employed a state-space model to decompose the strain into component responses and discovered the synchronous acceleration of approximate negative entropy anomalies at multiple stations four to six months before an earthquake. Liu et al., (2019) used the S-transform method to analyze the time-frequency characteristics of borehole strain data, revealing reliable anomalies that reflect the entire process of pre-earthquake, during, and post-earthquake strain changes. Chi, (2013) uncovered a "tidal aberration" phenomenon, persisting over three months before the earthquake, with significant strain

changes occurring 15 to 19 days prior. Tang and Jing, (2013) conducted an analysis of surface strain co-seismic orders, noting differences between the Wenchuan and Lushan earthquakes related to earthquake magnitude. Despite the valuable insights gained from these studies, they mostly focused on single-station data, overlooking the potential correlations between multiple stations. The study of seismic monitoring data based on multiple stations has been applied to many scenarios. Liu et al., (2019) analyzed the abnormal fluctuations of aerosol optical depth (AOD) before and after the 2008 Wenchuan

earthquake and the 2013 Lushan earthquake, and found that the abnormal high AOD values appeared 11 days before the Wenchuan earthquake and 4 days before the Lushan earthquake. It is considered that the AOD index may be suitable as a precursor to the earthquake in the Sichuan Basin. Using borehole strain data from six stations in the Sichuan-Yunnan region, Yu et al., (2020) established a graph network and analyzed 13 earthquake cases with $Es > 10^7$ in the study area. It was found that the strain anomaly before the earthquake generally occurred within the first 30 days of the earthquake event. To study

the abnormal strain changes before the Wenchuan earthquake, Zhu et al. (2019) introduced negative entropy analysis to the borehole data of three stations. The results show that Guza and Xiaomiao stations have similar trends and may record abnormal changes related to the Wenchuan earthquake. Renhe station failed to detect the anomalies before the earthquake due to the distance. An example of multi-station analysis is given, which shows that it is feasible to analyze seismic data with multi-station.

As earthquake monitoring data accumulates, traditional processing methods face challenges in managing vast quantities of data. The emergence of deep learning, particularly graph neural networks (GNNs), has markedly enhanced prediction and classification accuracy, particularly for non-Euclidean spatial data characterization (Kipf and Welling, 2016; Niepert et al., 2016; Scarselli et al., 2009; Wu et al., 2019; Zhou et al., 2020). Recent developments in spatiotemporal GNN frameworks integrate GNNs with various event-learning methods to extract complex dependencies (Oord et al., 2016; Rathore et al.,

2021; Yu et al., 2017). Kim et al., (2022) utilized raw waveform data from multiple stations to classify earthquake events, demonstrating the effectiveness of GNNs in aggregating features from individual stations. Bilal et al., (2022) refined earthquake magnitude, depth, and location predictions by extracting features from waveform data from multiple stations and integrating earthquake catalog information into a GNN. Huang et al., (2023) applied a Graph Attention Isomorphic Network (GAIN) to analyze earthquake catalogs and geomagnetic signals, successfully detecting pre-earthquake anomalies. These

studies highlighted the significant potential of GNNs in earthquake research.

    In this study, we proposed an innovative method for extracting pre-earthquake anomalies from borehole strain data using Graph WaveNet. The remained of this article is structured as follows: the next section delve into the specifics of the Lushan earthquake, providing an introductory exploration of the observation data pivotal to our analysis. In the third section, we introduce the SVMD and delve into the theoretical underpinnings of the graph wavenet network, laying the groundwork for a

comprehensive understanding of our analytical approach. A detailed case study of the Lushan earthquake follows, providing a tangible illustration that guides readers through the intricacies of our data-processing methodology. Section five mainly includes the analysis of prediction results, the detailed analysis of randomly selected abnormal days and the analysis of abnormal accumulation results. The sixth part is the discussion, which mainly includes the comparison and discussion of the abnormal accumulation results between different stations and the exclusion of the influence of meteorological factors. The

final section presents the conclusions of the study and summarizes the key insights drawn from our analysis.

## 2 Observation Data

Dobrovolsky's estimate of the radius of influence of precursors for earthquakes of different magnitudes is shown in the following Eq. (1) (Dobrovolsky et al., 1979):

$$\rho = 10^{0.43M} \text{ km} , \tag{1}$$

where $M$ denotes the magnitude and $\rho$ denotes the radius of influence of $M$ magnitude. The Lushan 7.0 magnitude earthquake's radius of influence extends approximately 1023 km. Among the selected monitoring stations, Guza is situated 73 km from the epicenter, while Xiaomiao, Luzhou, and Zhaotong are positioned at distances of 268 km, 286 km, and 337 km from the epicenter, respectively. This positioning confirms that the chosen stations possess the capability to monitor earthquake-related anomalies. Detailed information about Guza, Xiaomiao, Luzhou, and Zhaotong, including latitude and

longitude coordinates, distance from the Lushan earthquake's epicenter, rock type at the drill borehole locations, and borehole depth, is provided in Table 1. Figure 1 visually depicts the geographical locations of the four observation stations relative to the epicenter of the Lushan earthquake.

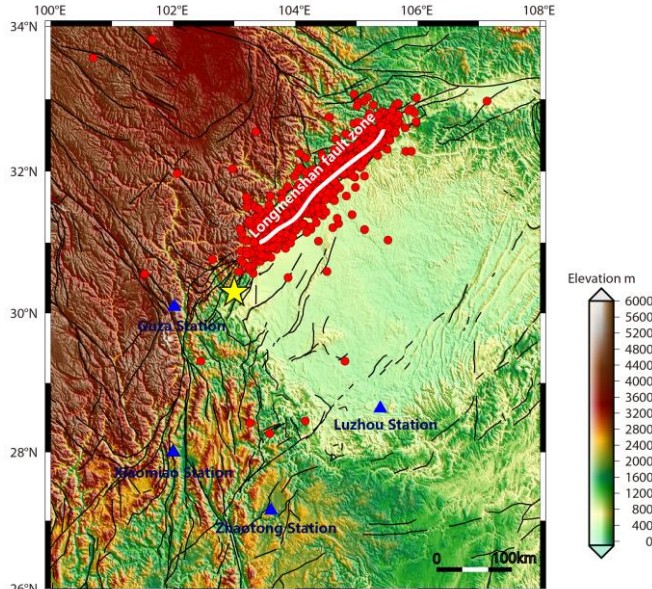

**Figure 1: Location of the four observation stations relative to the epicenter of the Lushan earthquake. The blue triangles represent**
**the locations of the borehole strain observation stations. The yellow star represents the epicenter of the Lushan earthquake, while the white curve depicts the Longmenshan fault zone. This map was generated by GMT software, v. 6.0.0rc5 (https://gmt-china.org/).**

**Table 1.** Information on borehole strain stations was used in this study.

| Station Name | Locations | Rock Type | Borehole Depth(m) | Epicentre Distance(km) |
|---|---|---|---|---|
| GuZa | 30.12° N,102.18° E | Granite | 40.69 | 73 |
| XiaoMiao | 28.00° N, 102.00° E | Siltstone | 41.78 | 268 |
| LuZhou | 28.87° N, 105.42° E | Quartz Sandstone | 40 | 286 |

| ZhaoTong | 27.32° N, 103.73° E | Basalt | 45 | 337 |

The four-component borehole strainmeter serves to observe the temporal inverse of displacement at a specific point, offering insights not attainable through GPS and seismometers. Operating with a continuous recording frequency of one sample per minute, it significantly enhances temporal resolution by at least one order of magnitude. Additionally, its observation bandwidth surpasses that of seismometers, particularly at the long-period end of the spectrum. This specialized YRY-4 strainmeter comprises four horizontally positioned sensors designed to measure changes in borehole diameter. These sensors are strategically spaced at a 45° angle, and the relationship between the four measurements from the strainmeter can be expressed as follows (Qiu et al., 2009; Su, 2019):

$$S_1 + S_3 = k(S_2 + S_4) , \tag{2}$$

This Eq. (2) represents the self-consistent formulation for a four-component borehole strain strainmeter. The self-consistent coefficient, denoted as $k$, ideally equals 1, and data is deemed reliable when $k$ is greater than or equal to 0.95. Strain conversion is achieved through the following Eq. (3) using the four measurements:

$$\begin{cases} S_{13} = S_1 - S_3 \\ S_{24} = S_2 - S_4 \\ S_a = (S_1 + S_2 + S_3 + S_4)/2 \end{cases} , \tag{3}$$

In this Eq. (3), $S_{13}$ and $S_{24}$ represent two independent shear strains. Shear strain pertains to alterations in the total area or volume of an object while maintaining a deformed shape. Additionally, $S_a$ denotes surface strain, signifying changes in area without a concurrent shift in the object's morphology. This characteristic is observed in the presence of hydrostatic enclosure pressure (Su, 2019). For the analysis in this paper, $S_a$ data from four stations—Guza, Xiaomiao, Luzhou, and Zhaotong— were specifically selected for examination.

Despite its advantages, such as high sensitivity and a wide frequency band during observation, the four-component borehole strainmeter remains susceptible to interference from surrounding sources. We used the improved VMD algorithm to analyze the $S_a$ data, and found that the first two components in the decomposition results correspond to the annual trend term and the solid tide, respectively, and the remaining components contain a large number of strain signals. We retained the remaining components as research data. Because there is no ability to extract meteorological factors such as air pressure, temperature and rainfall from the remaining components, we analyze the measured data of meteorological factors to determine whether the meteorological data affects the results of borehole strain observation.

**3 Methods**

To analyze borehole strain data from multiple stations for the Lushan earthquake, our study employed the flowchart shown in Fig. 2.

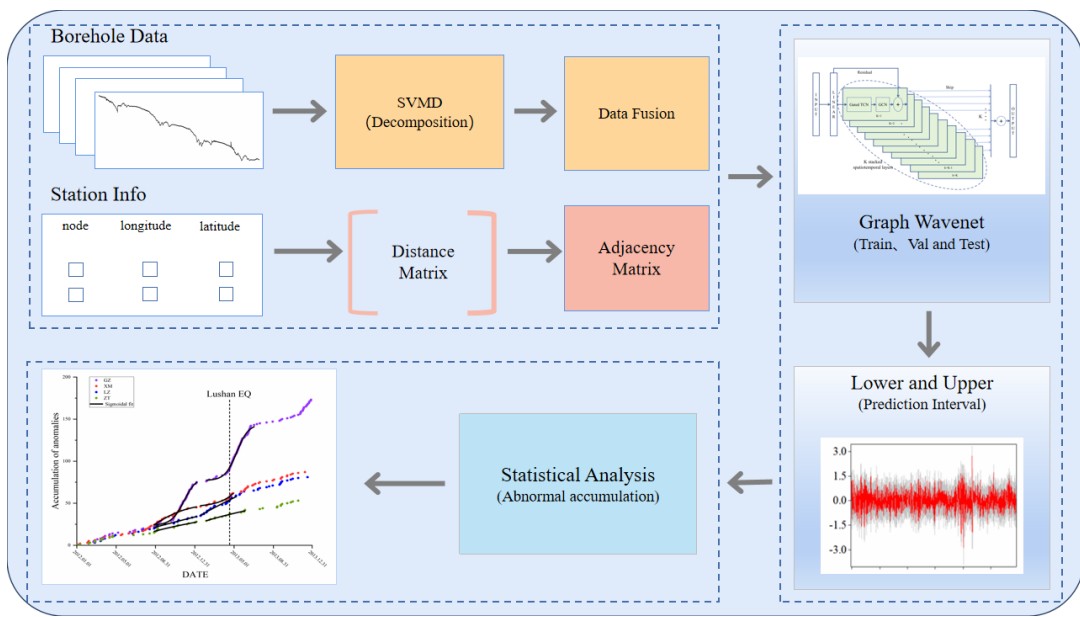

**Figure 2: The framework of borehole strain data processing and pre-earthquake anomaly detection.**

As shown in Fig. 2, the process begins with the conversion of data from the four-component borehole into strain. Subsequently, the borehole data are decomposed using the segmented variational modal method (SVMD), and the resulting decomposed outcomes were fused. By calculating distances between stations using longitudinal and latitudinal coordinates, a distance matrix is constructed and then normalized to form an adjacency matrix. The fused data and adjacency matrix serve as inputs for training, validation, and prediction using the graph wavenet GNN. During the prediction phase, upper and lower bound prediction intervals are established based on the model's output. Anomalies were identified by comparing the prediction intervals with the original data. The cumulative values of pre-earthquake anomalies in the borehole strain data from various stations were subsequently subjected to statistical analyses.

## 3.1 Segmented Variational Modal Decomposition

A Borehole Strain Signal, characterized as a typical nonstationary signal, can be effectively analyzed by decomposing it into a set of Intrinsic Mode Functions (IMFs) using the Empirical Mode Decomposition (EMD) method. However, EMD encounters challenges such as mode aliasing. To address this issue, an adaptive time-frequency analysis algorithm, the Variational Mode Decomposition (VMD), was introduced by (Dragomiretskiy and Zosso, 2014). The VMD exhibits superior noise immunity in signal processing, providing an effective solution to these challenges. This approach has been successfully applied by (Huang et al., 2022; Zhang and He, 2023) to process raw earthquake waveforms, yielding improved decomposition results.

VMD stands as a non-recursive signal processing method designed to decompose a time series into a sequence of intrinsic mode functions characterized by limited bandwidth. The decomposition process essentially involves solving variational problems, and the variational model can be expressed as follows:

$$\min_{\{u_k\},\{\omega_k\}} \sum_{k=1}^{K} \left\| \partial_t \left\{ \left[ \left( \delta(t) + \frac{j}{\pi t} \right) \bullet u_k(t) \right] e^{-j\omega_k t} \right\} \right\|_2^2$$

$$s.t. \sum_{k=1}^{K} u_k(t) = f(t) , \tag{4}$$

where $\{u_k\} = \{u_1, u_2, \dots, u_K\}$、$\{\omega_k\} = \{\omega_1, \omega_2, \dots, \omega_K\}$ are the $k$ modal functions and the corresponding center frequencies of the signal decomposition, respectively; $\partial_t$ is the bias computation for time $t$; $\delta(t)$ is the unit impulse function; $j$ is the imaginary unit; $\bullet$ is the convolution computation.

To solve the variational model, a quadratic penalty term $\alpha$ and the Lagrange multiplier operator $\lambda(t)$ are introduced to make the variational model unconstrained. The constructed generalized Lagrangian function is:

$$L(\{u_k\}, \{\omega_k\}, \lambda) = \alpha \sum_{k=1}^{K} \left\| \partial_t \{ [(\delta(t) + \frac{j}{\pi t}) * u_k(t)] e^{-j\omega_k t} \} \right\|_2^2$$

$$+ \|f(t) - \sum_{k=1}^{K} u_k(t)\|_2^2 + \langle \lambda(t), f(t) - \sum_{k=1}^{K} u_k(t) \rangle , \tag{5}$$

where $L$ denotes the Lagrangian generalization operator; $\alpha$ denotes the data fidelity constraint function; $\lambda$ denotes the Lagrangian multiplier.

The alternating direction multiplier method (ADMM) is used to solve Eq. (5), and the iterative optimization of $u_k$ , $\omega_k$ , and $\lambda$. The iterative formulas for the mode $u_k$ , the corresponding center frequency $\omega_k$ , and the Lagrange multiplier $\lambda$ can be updated as:

$$\hat{u}_k^{n+1}(\omega) = \frac{\hat{f}(\omega) - \sum_{i \neq k} \hat{u}_i(\omega) + \frac{\hat{\lambda}(\omega)}{2}}{1 + 2\alpha(\omega - \omega_k)^2}$$

$$\omega_k^{n+1} = \frac{\int_0^\infty \omega |\hat{u}_k(\omega)|^2 dw}{\int_0^\infty |\hat{u}_k(\omega)|^2 dw}$$

$$\hat{\lambda}^{n+1}(\omega) = \hat{\lambda}^n(\omega) + \tau(\hat{f}(\omega) - \sum_k \hat{u}_k^{n+1}(\omega)) , \tag{6}$$

where $\hat{u}_k^{n+1}$ is the $k$th IMF component in the n+1st iteration; $\omega_k^{n+1}$ is the center frequency corresponding to $\hat{u}_k^{n+1}$; $\hat{\lambda}^{n+1}(\omega)$ is the value of the Lagrangian operator in the n+1st iteration, and $\tau$ is the noise tolerance of the signal. Set the termination condition of the algorithm as:

$$\sum_{k=1}^{K} \frac{\|\hat{u}_k^{n+1} - \hat{u}_k^n\|_2^2}{\|\hat{u}_k^n\|_2^2} < \varepsilon , \tag{7}$$

where $\varepsilon$ denotes the discrimination accuracy.

In this study, we employ a method of applying Variational Mode Decomposition (VMD) to data segments through the incorporation of sliding windows (SVMD). This approach effectively addresses the challenge of limited memory when conducting VMD on the entire dataset while retaining the correlation between the data points (Chi et al., 2023). The fundamental principle of the SVMD method is illustrated in Fig. 3.

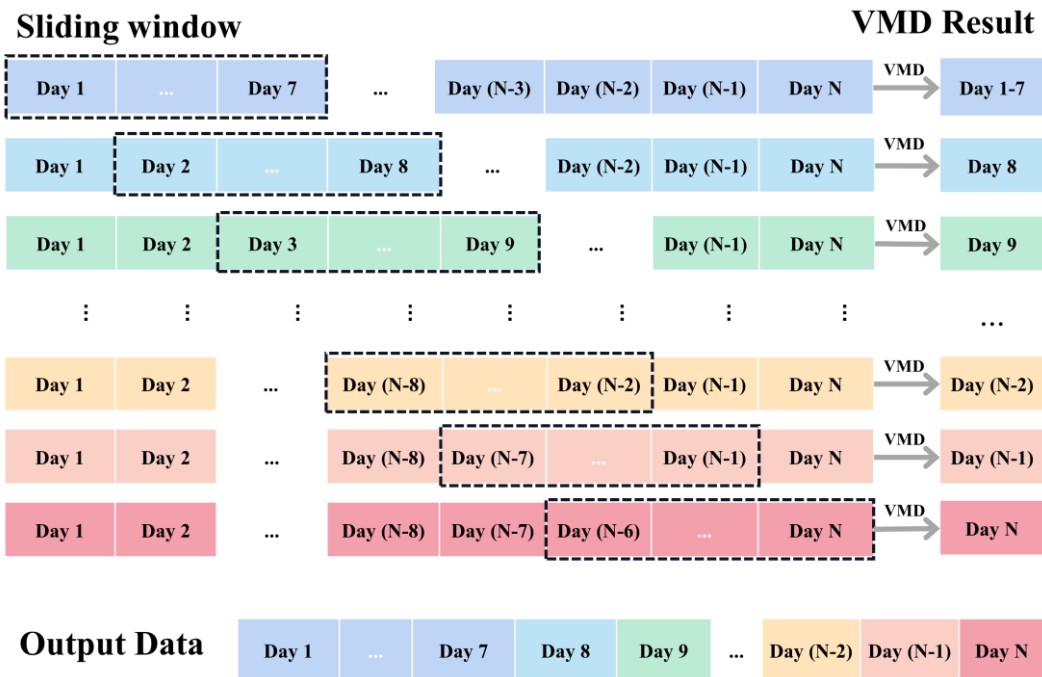

**Figure 3: Schematic diagram of Segmented Variational Modal Method (SVMD) principle.**

As depicted in Fig. 3, we opt for a consistent sliding window approach with a size of 7 days and a sliding step of 1. The initial sliding window encompasses all the data from the first 7 days. From the second sliding window onwards, only the data from the last day of the current window is preserved and concatenated behind the results obtained from the previous window.

### 3.2 Graph Wavenet Neural Network Architecture

### 3.2.1 Gated Temporal Convolutional Network for Extracting Temporal Features

During the processing of the time-series data, Causal Convolution maintains the causal relationships inherent within the data. This technique facilitates the extraction of time-series features through convolution. However, as the sequence length increases, capturing temporal dependence requires more convolution layers, thereby substantially increasing computational demands. To address this challenge, an expansion factor was introduced for Causal Convolution. The inclusion of this expansion factor can enlarge the receptive field of the Causal Convolution, enabling to capture longer time series features

with a reduced number of convolutional layers. The relationship between the input sequence length $L_{in}$ after causal dilation convolution and the output sequence length $L_{out}$ can be expressed as:

$$L_{out} = L_{in} + padding - (d \times (k - 1)),\qquad(8)$$

where $L_{out}$ is the length of the output sequence; $L_{in}$ is the length of the input sequence; padding is the number of zero paddings added to the ends of the input sequence, which are added at the beginning of the sequence to preserve causality; $k$

is the size of the convolution kernel, which is a small, learnable weight matrix; and $d$ is the dilation factor, which indicates

the span of the convolution kernel over the input sequence, with larger dilation factors being able to capture long-term time dependencies. The dilation causal convolution with a convolution kernel size of 2 and a sliding step of 1 is shown in Fig. 4a.

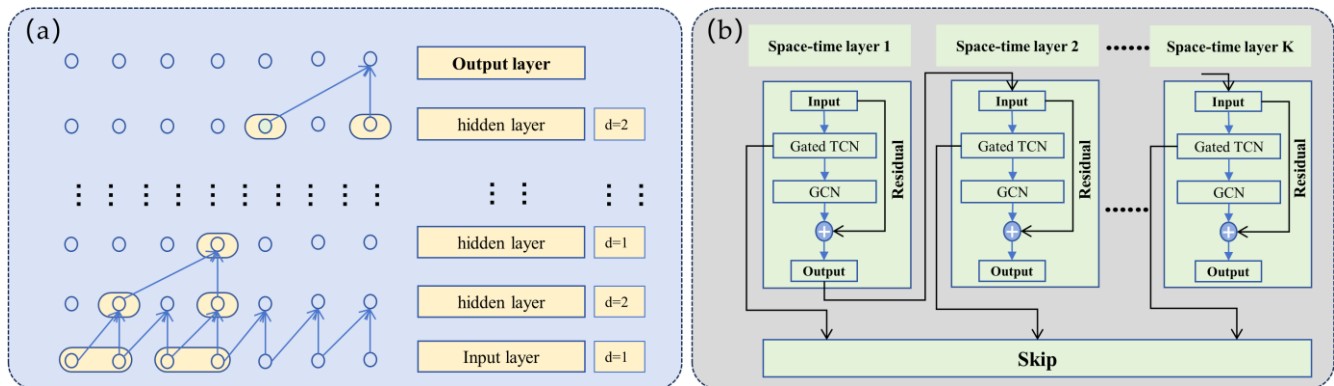

Figure 4: (a) Schematic of Dilated Causal Convolutional Layers. (b) Temporal-Spatial Layers architecture.

Although Convolutional Neural Networks (CNNs) are commonly employed for image processing, 1D CNNs can also be effectively used for time-series analysis. Gated Temporal Convolutional Networks (TCN) leverage 1D CNNs to extract features from time-series data. This architecture comprises two key modules: one for convolving the input to extract features by employing tanh as the activation function, and the other for controlling the amount of information passed from the current layer to the next layer by utilizing the sigmoid activation function. The Gated TCN module is defined as:

$T = \boldsymbol{g}(W_1 * x + b_1) \bullet \boldsymbol{\sigma}(W_2 * x + b_2)\,,$                (9)

where $W_1$ and $W_2$ represent the weight parameters, $b_1$ and $b_2$ represent the corresponding bias parameters, and $\bullet$ denotes convolution, where $\boldsymbol{g}$ is the activation function of the output, $\boldsymbol{\sigma}$ is the activation function that determines the ratio of information passed to the next layer.

### 3.2.2 Graph Convolutional Networks for Extracting Spatial Features

The fundamental concept of a graph network is to represent interactions among real features based on spatial dependence dictated by the graph structure. The distance adjacency matrix is symmetrically normalized to function as an adjacency matrix for graph convolution. This approach effectively captures the node information and preserves the graph structure. By constructing an undirected complete graph for each station, where all nodes are directly connected, each node interacts solely with its neighboring nodes for information exchange. To ensure that input features encompassed all information in the
current node and its first-order nearest neighbor nodes, a first-order symmetric normalized adjacency matrix was selected during the graph convolution process. The convolution layer of the graph is defined as:

$$H^{(l+1)} = \sigma(\widetilde{D}^{-\frac{1}{2}} \cdot \tilde{A} \cdot \widetilde{D}^{-\frac{1}{2}} \cdot H^{(l)} \cdot W^{(l)})\,,\qquad\qquad(10)$$

where $H^{(l)}$ denotes the embedding vector of layer l; $H^{(l+1)}$ denotes the embedding vector of layer $(l + 1)$; $\widetilde{D}^{-\frac{1}{2}} \cdot \tilde{A} \cdot \widetilde{D}^{-\frac{1}{2}}$ denotes the symmetric normalized adjacency matrix of the current layer; where $D$ denotes the degree matrix and $\tilde{A}$ denotes

the Distance Adjacency Matrix; $W^{(l)}$ denotes the weight of the neural network in the current convolutional layer; and $\sigma$ denotes the nonlinear activation function of the neural network. Introducing a dropout rate to the output of each training batch in graph convolution involves randomly ignoring half of the hidden layer nodes.

This dropout strategy was applied across different neural networks, and "opposite" fits were averaged. This aids in mitigating overfitting because conflicting tendencies can cancel each other out during the training process.

### 3.2.3 Temporal-Spatial Layers


Figure 4b illustrates the structure of the spatiotemporal layer. Each layer within the spatiotemporal layer captures temporal dependencies using Gated TCNs. This involved utilizing node features produced by the TCN module and the graph's adjacency matrix as inputs for the Graph Convolutional Network (GCN) module. The GCN layer is responsible for capturing spatial-temporal features, and the spatial-temporal features of the current layer are residual and linked to the input, serving as

inputs to the subsequent spatial-temporal layer. The output of the Gated TCN module serves as the output of the current spatial-temporal layer, and the outputs of all spatial-temporal layers are interconnected using skip connections. This design facilitated the capture of short-term and spatial dependencies.

### 3.2.4 Graph Wavenet Neural Network Framework

Figure 5 illustrates the framework of a graph wavenet. The initial step involved pre-processing the multistation borehole data

and converting the latitude and longitude of each station into an adjacency matrix. Subsequently, the data underwent a dimensionality increase in the linear layer, followed by processing in the Gated TCN module to obtain the current information. The output of the Gated TCN module is then employed to extract spatial information through the GCN layer. The extracted spatial and temporal information is connected to the input of the current layer, which serves as the input for the subsequent spatial and temporal layers, which can capture both the current and historical information using different dilation

factor sizes. Finally, the information extracted from each Gated TCN layer is aggregated into an output layer. The output sequence was downscaled using the ReLU activation function and a linear layer. The upper and lower bounds of the output sequence were calculated using the normal distribution method. The prediction intervals of the network were constructed from the upper and lower bounds. The upper and lower bounds of the prediction intervals are determined using the following formulas:

$Lower = Prediction + Z \times \mathrm{rmse}$

$Upper = Prediction - Z \times \mathrm{rmse}$ ,                              (11)

where $Prediction$ is the predicted value; $Z$ is the Z-Score of the normal distribution, which is about 1.96 for a 95% confidence level; and $rmse$ is the root mean square error.

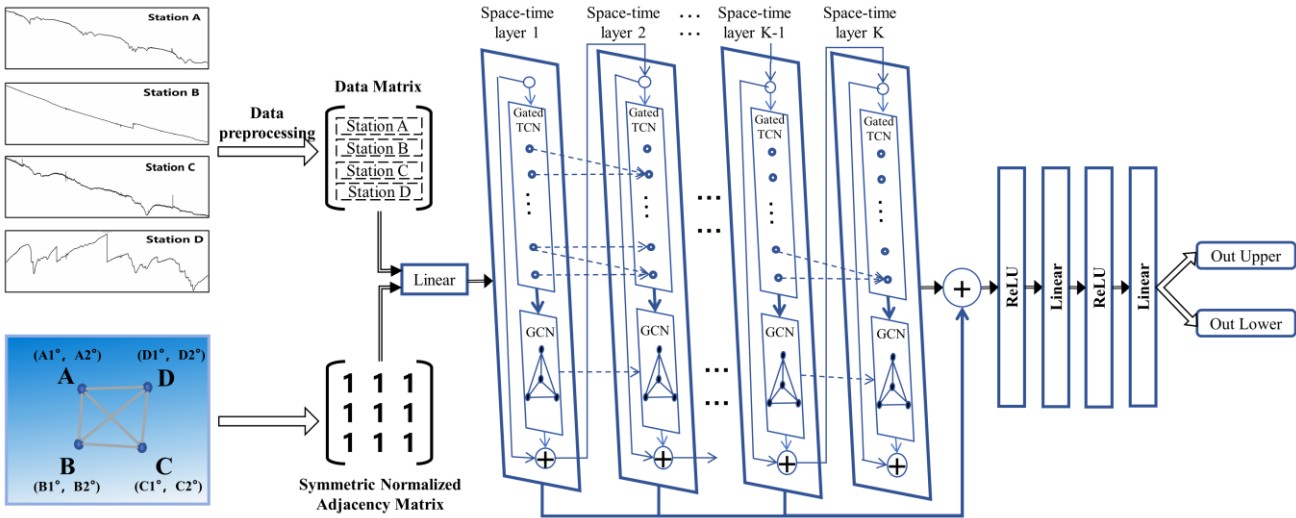

**Figure 5: Framework of Graph Wavenet.**

For the graph wavenet neural network model, Mean Absolute Error (MAE) was employed as the loss function for backpropagation during training. A dropout of 0.3 was applied during graph convolution to enhance the model's generalization. The Adam optimizer was utilized to update the weights, with a learning rate of 0.001 and a weight decay rate of 0.0001. This configuration allowed the model to decay, effectively preventing over-fitting by reducing parameter

magnitudes. The number of training rounds for the model was set to 100.

## 4 Data Processing

We analyzed the four-component borehole strain data collected from Guza, Xiaomiao, Luzhou, and Zhaotong stations from 1 January 2010 to 31 December 2013. Initially, the data from each station were validated using self-consistent equations. Subsequently, the four-component borehole strain data from each station were transformed into two shear strains, $S_{13}$ and $S_{24}$,

and one surface strain, $S_a$, using a strain conversion equation. Figure 6a shows the data series of surface strain $S_a$ after converting the borehole strain data from each of the four stations.

Subsequently, the $S_a$ data from each of the four stations were decomposed using the Segmented Variational Modal Method (SVMD). The decomposition parameters were set to a bandwidth of 2,000, the number of modes decomposed to five, and the convergence accuracy to $10^{-7}$. The results of the SVMD decomposition were compared with data related to the

influencing factors, effectively eliminating the effects of seasonal trends and solid tides. The extracted modes are specifically related to crustal activity, showcasing short-period, high-frequency oscillatory signals associated with earthquake variations in earthquake strain within crustal motions (Chi et al., 2019). The time-series data after SVMD decomposition of $S_a$ data from each station are illustrated in Fig. 6b.

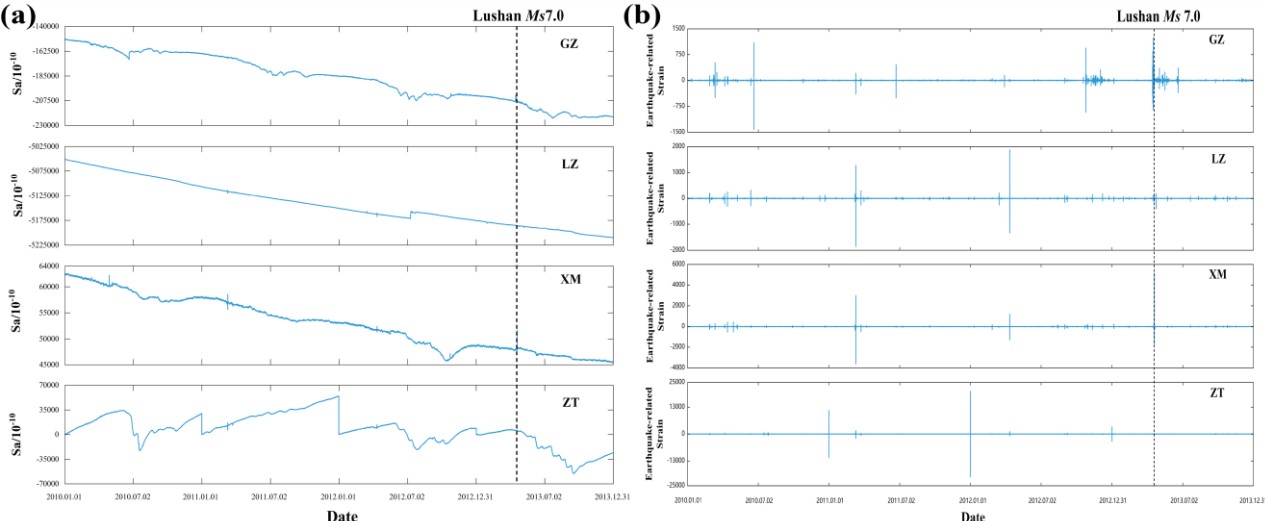

Figure 6: (a) $S_a$ of the borehole strain data from 2010 to 2013 for the Guza, Xiaomiao, Luzhou, and Zhaotong stations. (b) SVMD results of $S_a$ components for Guza, Xiaomiao, Luzhou and Zhaotong stations.

The next step involved data fusion, with SVMD applied to the $S_a$ component data from different stations to extract data related to crustal activities. The construction of the GNN required information from each node; hence, the information from multiple stations was fused as inputs to the GNN, and the data matrix $R_a$ of the constructed $R_a$ component is shown below:

$$R_a = \begin{bmatrix} GZ_a(t_1) & GZ_a(t_2) & ... & GZ_a(t_{2103840}) \\ LZ_a(t_1) & LZ_a(t_2) & ... & LZ_a(t_{2103840}) \\ XM_a(t_1) & XM_a(t_2) & ... & XM_a(t_{2103840}) \\ ZT_a(t_1) & ZT_a(t_2) & ... & ZT_a(t_{2103840}) \end{bmatrix}^T, \tag{12}$$

Where GZ , LZ , XM , and ZT represent the Guza, Luzhou, Xiaomiao, and Zhaotong stations, respectively; $a$ indicates the selection of the surface strain $S_a$ component data; $t$ denotes the length of the time series; and $T$ denotes the transpose; and the fused data matrix $R$ serves as the input for the node information in the Graph Wavenet GNN. The constructed data matrix has a minute sampling interval with 1,440 data sampling points per day. The data matrix undergoes processing using a sliding window with a length of 60, where the sliding window size of 60 corresponds to one hour. A sliding window was implemented to predict the next hour's data based on the current hour's data. The sampled data are considered as the features of the sample, and the data shape is constructed as a tensor of [64, 60, 4, 1], where 64 denotes the sample length and [60, 4, 1] represents the input or output of a single data point. In this context, 60 signifies the sequence length, 4 is the number of nodes, and 1 is the number of node features. The selected data spanned 2010 to 2013, divided chronologically in a ratio of 3:1:4. Specifically, 2010 and 2011 were allocated as the training and validation sets, respectively, whereas 2012 and 2013 were used as the test set.

The subsequent step involved constructing the adjacency matrix of the graph. A node graph of the four stations is shown in Fig. 7. The graph is defined as $G = (V, E)$, where $V$ corresponds to the set of nodes and $E$ corresponds to the set of edges in the graph. The relationship between edges and nodes is expressed as $E_{ij} = (V_i, V_j), V_i, V_j \in V$. Graphs are commonly

represented using an adjacency matrix, where the adjacency matrix $A$ is a $N \times N$ square and $N$ denotes the number of nodes. The graph is written as $Aij = 1$ if two vertices, $i$ and $j$, are connected by an edge and 0 otherwise. The number of neighboring nodes for node $V$ is referred to as the degree of that node, and the degree matrix $D$ is an $N \times N$ diagonal array, with the elements on the diagonal being the degrees of individual vertices: $D(V_i) = N(i)$.

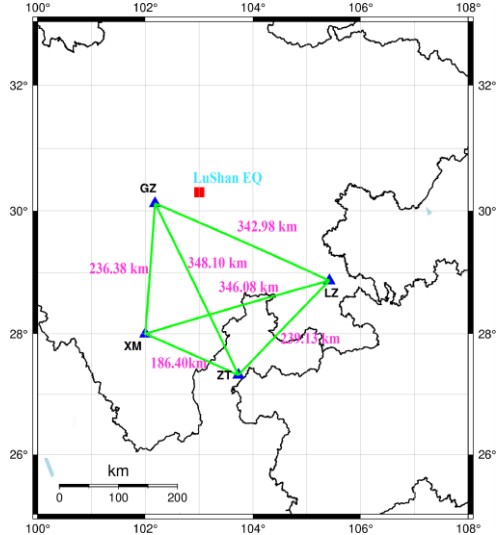

**Figure 7: Node diagram of the four borehole strain observation stations constructed. The blue triangles indicate the locations of the four borehole strain stations, the green line indicates the distance between the two stations, and the red square indicates the epicenter of the Lushan earthquake.**

The true distance between the stations was calculated using the corresponding latitude and longitude data of any two stations. Assuming that the Earth is a standard sphere, the principle for calculating the distance in latitude and longitude 310  coordinates involves determining the distance between two points on the sphere, equivalent to the arc length of a cross-sectional circle. For any two given points and their corresponding latitudes and longitudes, $A(N1, E1)$ and $B(N2, E2)$, with the average radius of the Earth as $R$ and the center of the Earth as the midpoint of the right angle coordinates, $A$ and $B$ represent two points corresponding to the right angle coordinates:

$A(R \cos(N1) \cos(E1)), R \cos(N1) \sin(E1), R \sin(N1))$,$A(x_1, y_1, z_1)$

$B(R \cos(N2) \cos(E2)), R \cos(N2) \sin(E2), R \sin(N2))$,$B(x_2, y_2, z_2)$ ,                    (13)

We calculated the angle between two points based on the coordinates of points $A$ and $B$. Let the angle between $A$ and $B$ be $\alpha$; then, the cosine of the angle $\cos\alpha$ is calculated as follows:

$$\cos\alpha = \frac{x_1 * x_2 + y_1 * y_2 + z_1 * z_2}{\sqrt{(x_1^2 + y_1^2 + z_1^2)} * \sqrt{(x_2^2 + y_2^2 + z_2^2)}} ,$$                    (14)

Thus, the distance $d$ between two points can be expressed as:

$d = $ R$*arccos(cos\alpha)$ ,                    (15)

The distance between two stations was determined by calculating the latitude and longitude coordinates of any two stations. Subsequently, an adjacency distance matrix for the node graph is constructed based on these distances. To optimize the adjacency matrix for use in the Graph Wavenet GNN, the distances between the nodes were normalized to represent the weights between them. This normalized adjacency matrix was then utilized in the Graph Wavenet model.

**5 Results**

In this study, we employed a Graph Wavenet GNN to analyze borehole data from multiple stations prior to the Lushan earthquake. The analysis focused on extracting pre-earthquake anomalies based on the results obtained. Anomalies were identified when the raw data surpassed the corresponding upper or lower prediction intervals established by the network. The prediction results for each station are shown in Fig. 8.

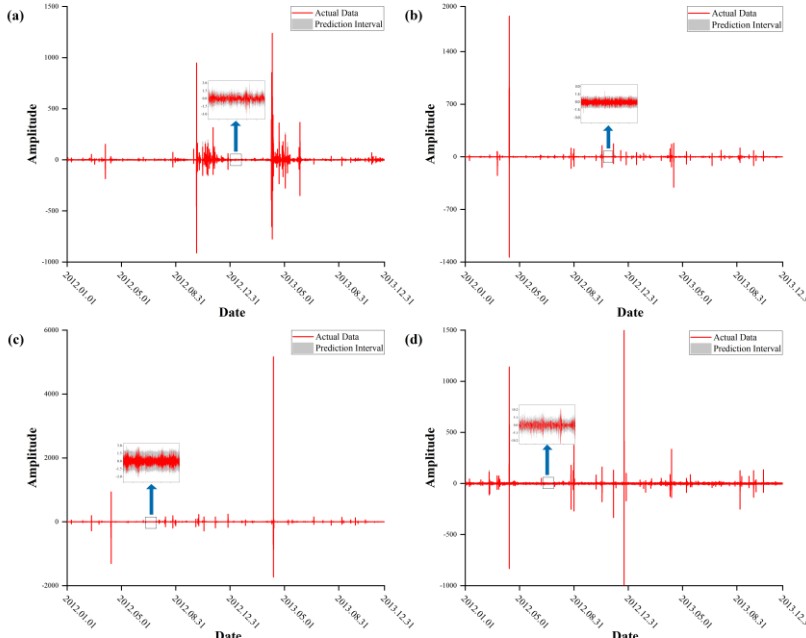


**Figure 8: Graph WaveNet graph neural network prediction results; red lines indicate real data and gray areas indicate prediction intervals. (a) Results of the Lushan earthquake prediction from Guza station; (b) Results of the Lushan earthquake prediction from Luzhou station; (c) Results of the Lushan earthquake prediction from Xiaomiao station; (d) Results of the Lushan earthquake prediction from Zhaotong station.**

As shown in Fig. 8, our raw data closely align with the predicted intervals, demonstrating the Graph Wavenet accurate prediction of borehole data at each station. To identify point anomalies in the borehole strain data predictions, we employed the following criteria: (a) detecting more than 15 points outside the intervals within a 30-minute window; (b) identifying difference between the center of predicted intervals and the actual values exceeding 1.5 times the bandwidth of the intervals, with more than three such points in the same 30-minute period. Days meeting these conditions were considered anomalous

(Chi et al., 2023). To validate whether the extracted anomalous days were earthquake-related, we randomly selected raw data for four anomalous days from each station, as shown in Fig. 9.

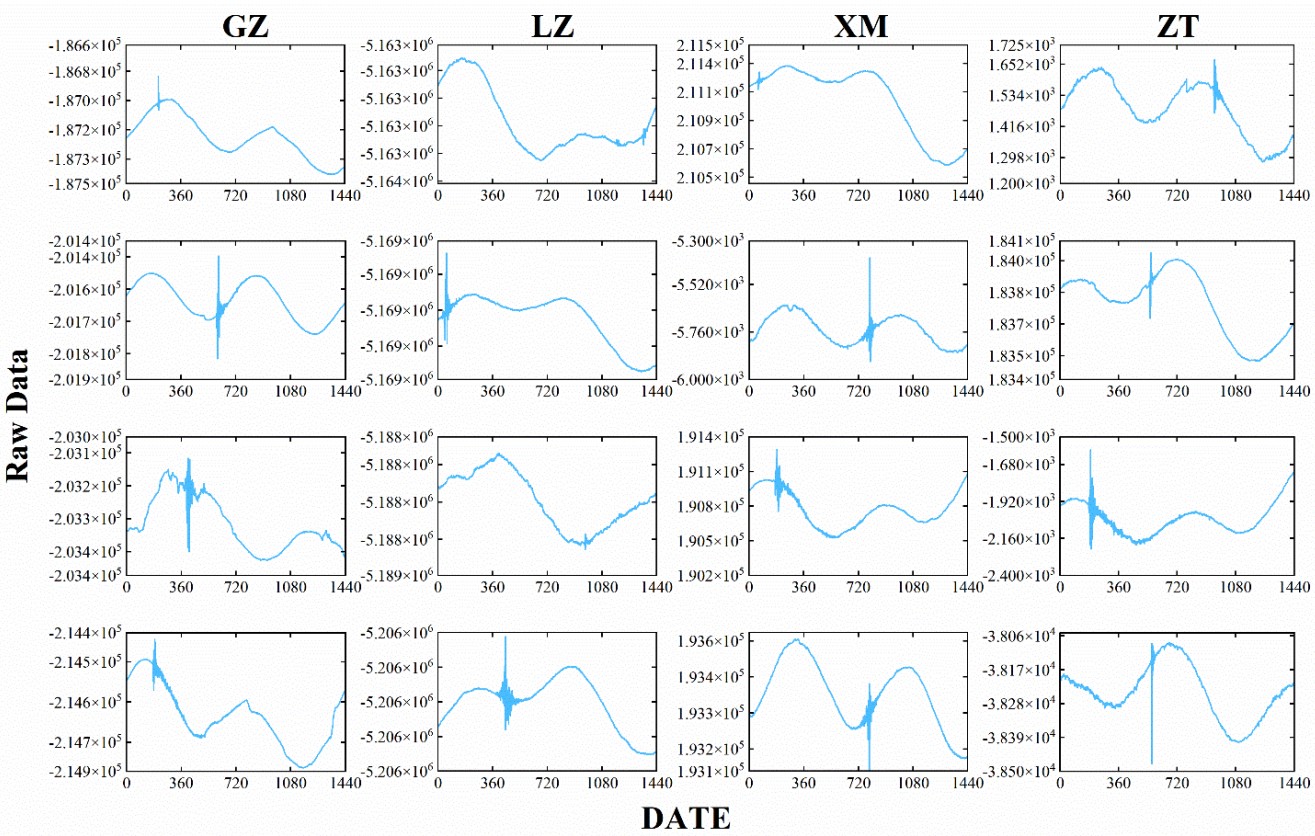

**Figure 9: Plots of raw data from four randomly selected anomalous days at each station.**

  In Fig. 9, it is evident that the abnormal days we defined exhibit short-period, high-frequency oscillation signals in the
original waveform, suggesting that these days are associated with crustal activity. De Santis et al., (2017) study the 2015 Nepal event using Swarm magnetic satellite data. For the first time, an S-shaped fitting function was proposed in the abnormal accumulation analysis, and some abnormal differences were found in the area around the EQ epicenter from the abnormal accumulation results. By comparing the S-shaped function and the linear fitting, it was found that the S-shaped fitting was significantly better than the linear fitting. In this paper, the S-type function is used to fit the abnormal
accumulation results. The cumulative values of the anomalous days over time are depicted in Fig. 10.

  As depicted in Fig. 10a, the cumulative results of the anomalous days at Guza station exhibit a two-part concavity. One part displayed a rapid increase in the number of anomalous days from October 2012 to three months before the earthquake (January 2013), after which it leveled off. Our findings align with those of (Chi et al., 2013) and (Zhu et al., 2018), indicating that strain anomalies occurred during the 4-8 months preceding the earthquake, and the accumulation of
anomalous days exhibited an accelerating trend. In the other segment, the number of anomalous days started to rise sharply

from March 2013 to June after the earthquake, after which the increase in anomalous days gradually leveled off. Relevant researchers have studied the short-term anomalies before the upcoming Lushan earthquake and believe that these anomalies occurred within a few days to a month before the earthquake(An et al., 2013; Jiang et al., 2013; Yu et al., 2020; Qiu et al., 2013; Zhu et al., 2018). Our study yielded similar results, indicating anomalies in the one-month pre-earthquake period of the Lushan earthquake. Therefore, we posit that these anomalies might be related to the Lushan earthquake. The findings of (Yu et al., 2021), who observed a brief increase in anomalies followed by a return to a steady state in the two-month post-earthquake period, align closely with the outcomes of our study. Li et al., (2017) analyzed the pre-earthquake anomalies of the Lushan earthquake using seismic rate data. The synthesized results show that there is an anomaly of rising earthquake frequency over time that lasts for 3-5 months from September 2010, which is highly consistent with our extracted anomaly of rapidly increasing anomalous days about 6 months before the earthquake. It indicates that we extracted strain short-term precursor anomalies associated with the Lushan earthquake.

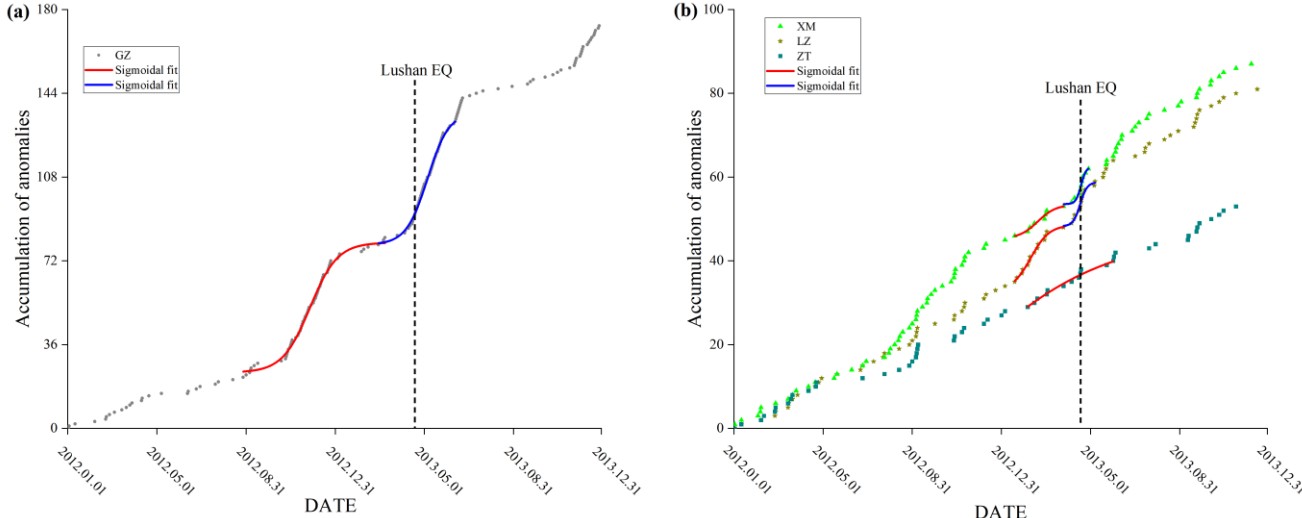

**Figure 10: Accumulation results of anomalous days in borehole data from each station. The dashed line indicates the date of the earthquake, while the red and blue curves indicate the results of the S-shaped fitting before and after the earthquake, respectively. (a) Anomalous day accumulation results for Guza station; (b) Anomalous day accumulation results for Xiaomiao, Luzhou, and Zhaotong stations.**

The findings of this study align with the theory of the synergism process of a fault. Ma and Guo, (2014) conducted a laboratory modeling study on the instability of a planar strike-slip fault, suggesting that the occurrence of an earthquake is linked to a fault's synergistic process, which encompasses three stages. In the initial stage, there's a deviation of the stress curve from linearity. The second stage is marked by the steady increase and expansion of isolated areas of strain release. In the final stage, the fault's sections of strain release accelerate and expand, alongside a rapid increase in strain levels in areas of strain accumulation. The period from September to December 2012 corresponds to the first and the second stages, where the stress curve deviates from linearity and isolated areas of strain release grow and extend steadily. From early 2013 up to the earthquake, aligns with the third stage, characterized by the accelerated expansion of strain release sections on the fault

and a swift rise in strain levels in strain-accumulation areas. The multitude of anomalies observed post-earthquake, including those caused by crustal fractures and aftershocks, were also evident. Similar phenomena were recorded at the XM and LZ stations, correlating with Ma's theory. Thus, we believe that the anomalous phenomena observed prior to the Lushan earthquake are related to the earthquake's gestation process.

    Xu et al., (2019) used GNSS observation network data to study the deformation before the Lushan earthquake and found
that a locking state that lasted from 5 months before the earthquake to 2 months after the earthquake occurred before the Lushan earthquake, and that under the locking state, the strain energy was still accumulating until the fault ruptured. When the stress accumulation in the pregnant seismic zone enters the nonlinear accumulation stage from linear accumulation, the resulting stress perturbation will lead to changes in the additional stress and strain states at nearby strain measurement points as the degree of stress and strain accumulation is further enhanced. Rock rupture experiments and theoretical studies have
shown that pre-slip occurs before fault stick-slip, and the resulting stresses can lead to obvious anomalous responses at nearby stations (Li, 2002; Ma et al., 1998; Zhao et al., 1997). Zhang et al., (2020) conducted an analysis of cross-fault deformation preceding the Lushan Ms7.0 earthquake. Their findings revealed a considerable shift in the cross-fault deformation dynamics of the study area, transitioning from a state of "inheritance," conducive to stress accumulation, to "reverse inheritance" more than six months prior to the earthquake. This shift exhibits characteristics of coordinated and
accelerated fault activities, aligning with the metastable and sub-unstable states observed in the structural mechanics tests. We believe that even in a locked state, energy pre-release still occurs in the locked part of the fracture. When the strength of rock rupture enters the destabilization stage, the way of stressing the adjacent rocks before rupture shows obvious tension and compression regions, the reason for the generation is related to the local extension and weakening, and most of the anomalies will appear in the form of sudden jumps. In summary, our analysis suggests that 4-8 months prior to the Lushan
Ms7.0 earthquake, the southern section of the Longmenshan rupture exhibited characteristics of a sub-stabilized state. This state led to the formation of relatively weak segments on the fault, contributing to an increase in the number of anomalous days potentially associated with the pre-release of energy. Simultaneously, the relatively strong fault segments underwent strain accumulation. In the days immediately preceding the earthquake, these strong segments reached a destabilized state owing to the accumulated strain, ultimately facilitating the occurrence of an earthquake (Ma and Guo, 2014).

**6 Discussion**

In Fig. 10b, we analyzed the anomalous day accumulation at other stations within the graph wavenet network. Remarkably, the accumulation patterns of anomalous days at Xiaomiao and Luzhou closely mirrored those observed at Guza, featuring concave trends in two distinct phases. However, the fitting results for the Zhaotong station deviated from the observed pattern. Analyzing the aforementioned fitting results, we observed a similar trend between the Xiaomiao and Luzhou stations,
resembling the patterns identified at the Guza station. The initial phase witnessed a sharp increase in January 2013 and plateaued until March 2013, indicating an accelerated accumulation of abnormal days in the four months leading up to the

earthquake. In the second phase, spanning from 5 April 2013 to 28 April 2013, a surge was apparent 15 days prior to the earthquake, followed by a gradual decline 8 days post-earthquake. This pattern echoes the impending and post-earthquake anomalies observed at the Guza Station during the Lushan earthquake. A detailed analysis indicates that the Xiaomiao and Luzhou stations also detected anomalous signals during the gestation process of the Lushan earthquake, affirming that the anomalies at these three stations were not random but were indeed linked to the Lushan earthquake. Further scrutiny of the distances between the stations and the epicenter revealed a noteworthy pattern: Guza station, which was the closest to the epicenter, recorded the highest number of anomalous days; Xiaomiao and Luzhou stations, in proximity to the epicenter, registered fewer anomalous days than Guza station, whereas Zhaotong station, situated farther from the epicenter, reported the fewest anomalous days. This lends credence to the belief that the anomalous signals received by a station are associated with the distance between the station and epicenter.

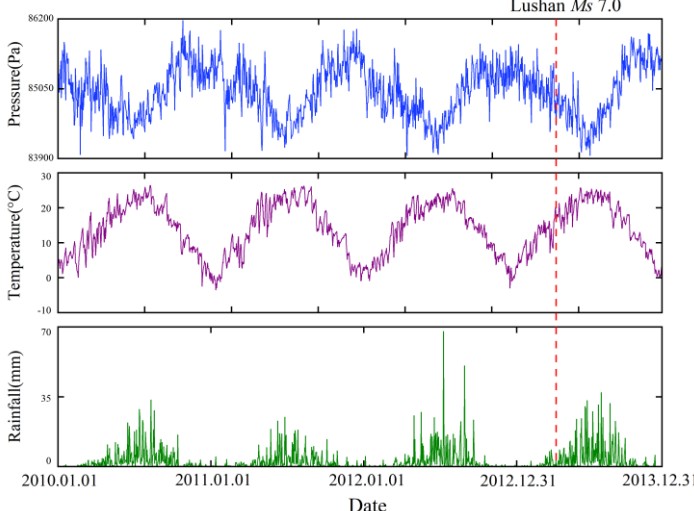

**Figure 11: Regional daily mean variations in pressure, temperature, and rainfall in the Lushan area from 2010 to 2013.**

Despite its advantages, such as high sensitivity and a wide frequency band during observation, the four-component borehole strainmeter remains susceptible to interference from surrounding sources. Figure 11 illustrates the impact of external factors by examining the regional daily mean data of pressure, temperature, and rainfall in the Lushan region (102° E, 27° N, 106° E, and 31° N), downloaded from 1 January 2010 to 31 December 2013, via NASA's Giovanni-4 platform (https://giovanni.gsfc.nasa.gov/giovanni). The analysis of these data revealed distinct annual trends in pressure, temperature, and rainfall. Both pressure and temperature exhibited fluctuations within a certain range, displaying opposite trends, whereas rainfall underwent a consistent increase followed by a decrease each year, in accordance with seasonal changes. To mitigate the impact of external factors on borehole strain data, we conducted a differencing process on the daily regional averages for pressure, temperature, and rainfall in the Lushan area. The periodic changes can be removed by differential processing, which highlights the anomaly of the data. The results are shown in Fig. 12.

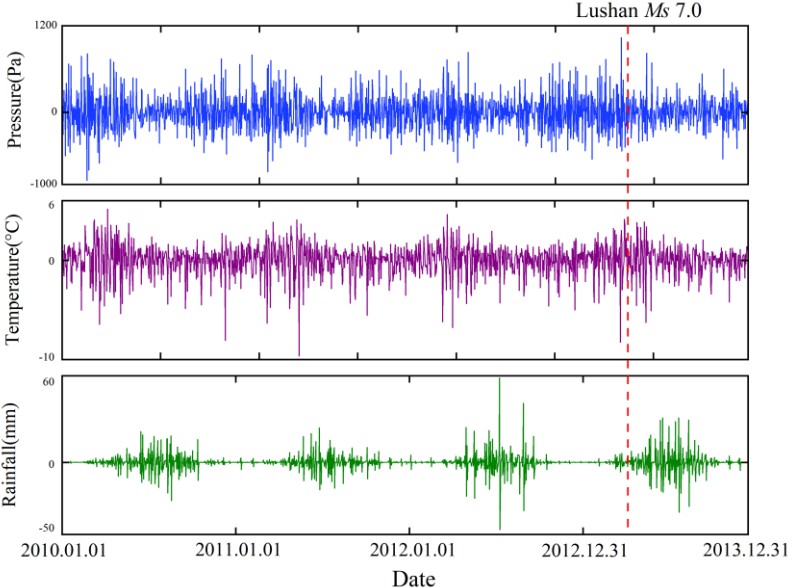

435 **Figure 12: Differing results of regional daily mean pressure, temperature, and rainfall in the Lushan area from 2010 to 2013.**

Figure 12 indicates that the regional daily mean differences in pressure, temperature, and rainfall in the Lushan area did not exhibit any anomalous changes. Therefore, we can exclude the influence of pressure, temperature, and rainfall on the anomalies observed in the pre-earthquake borehole data from Lushan. We have reason to believe that the anomalies we extracted before the Lushan earthquake are related to the seismogenic process.

## 7 Conclusion

In this study, we proposed a novel pre-earthquake anomaly extraction method based on a graph wavenet network structure that enables the integration of borehole strain data from multiple stations and makes predictions by learning both temporal and spatial correlations. The statistical analysis of pre-earthquake anomalies in borehole strain data from four stations, Guza, Xiaomiao, Luzhou, and Zhaotong, revealed two S-shaped upward trends in the pre-earthquake period for Guza, Xiaomiao, and Luzhou. This indicates that these three stations experienced notable strain anomalies during the gestational period of the Lushan earthquake. A comparison of anomaly accumulation rates among different stations indicated that the anomaly rate at Guza station was substantially higher than that at Xiaomiao and Luzhou stations, suggesting a correlation with distance from the epicenter. Raw data analysis of randomly selected anomalous days from each station confirmed the correlation between the extracted anomalous days and the pre-earthquake anomalies. Additionally, we analyzed regional daily averages of meteorological factors, preliminarily excluding their influence on the anomaly accumulation results. Therefore, we conclude that the graph wavenet network effectively extracted pre-earthquake anomalies from borehole strain data, highlighting its potential as a robust approach for studying pre-earthquake anomalies across multiple stations.

*Data availability*. The data that support the findings of this study are available from the China Earthquake Networks Center, but restrictions apply to the availability of these data, which were used under license for the current study, so are not publicly available. Data are however available from the Corresponding author (Email: chicqhainnu@gmail.com) upon reasonable request and with permission of the China Earthquake Networks Center.

*Author Contributions*. Conceptualization, Chenyang Li and Chengquan Chi; Data curation, Chenyang Li, Chengquan Chi and Yu Duan; Formal analysis, Chenyang Li, Chengquan Chi, Ying Han, and Zining Yu; Investigation, Chenyang Li; Methodology, Chengquan Chi; Resources, Chengquan Chi; Software, Chenyang Li and Dewang Zhang; Supervision, Chengquan Chi, Zining Yu and Ying Han; Validation, Chenyang Li, Chengquan Chi, and Ying Han; Writing – original draft, Chengquan Chi and Ying Han; Writing – review & editing, Chenyang Li and Chengquan Chi.

*Competing interests*. The authors declare that they have no conflict of interest.

*Acknowledgments*. The authors would like to thank Qiu Z. H., Tang L., and Yang D. H. from the China Earthquake Administration for giving essential help in accessing the website and downloading the strain data. The authors are also grateful to the Giovanni for meteorological data(https://giovanni.gsfc.nasa.gov/giovanni). The authors are also grateful to the China Earthquake Networks Center for borehole strain data.

*Financial support*. This work was supported by the Hainan Provincial Natural Science Foundation of China under Grants 621QN242,622RC669, 621QN0888, 322RC659 and 320QN253, as well as by the Program of Hainan Association for Science and Technology Plans to Youth R & D Innovation under Grants: QCXM202006; This work is supported by the Fundamental Research Funds for the Central Universities under Grants 202213042 and Youth Fund of the National Natural Science Foundation of China, project number:42204005.

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
