# Peer review of "Extraction of Pre-earthquake Anomalies in Borehole Strain Data Using Graph WaveNet: A Case Study of the 2013 Lushan Earthquake, China"

_EGUsphere, 2023_

## Referee Comment (RC2)

**General Comment**

The paper "*Extraction of Pre-earthquake Anomalies in Borehole Strain Data Using Graph WaveNet: A Case Study of the Lushan Earthquake*" presents a Graph Wavenet method to analyze the large amount of data collected at four borehole strainmeters before and after the Lushan Ms 7 earthquake. The authors highlight an acceleration of anomalies accumulation, and they infer a possible release of energy from a weak fault section and a strain accumultion on a strong section of the fault.
Although this work proposes a promising approach to analyze large chunks of data, two main drawbacks emerge in this reviewer opinion: (i) the paper does not flow optimally and it would need a better re-organization (see specific comments); (ii) it seems like the authors focused on the Graph Wavenet sometimes leaving behind a more accurate physical interpretation of the results obtained. As a suggestion, after having recognized the anomalous days, it would be interesting to have a more detailed discussion of these "anomalous data". Have you compared your results with different data (e.g., seismicity rates, pore-pressure data, deformations from GNSS measurements…)?

**Specific Comments:**

-Line 15: "acceleration of anomalous accumulation" of what? And in general I believe that the term "anomaly" should be better defined at the beginning of the paper.

-Line 48: which type of borehole strainmeters have been installed? Please specify…

-Lines 73-85: maybe this is too premature here? There is no need to go into the details of the algorithm here. I'd re-organize the introduction moving the description of the earthquake (lines 95-100) before the state of art (before line 28). I'd also be more coincise on the description of previous works on this earthquake.

-Lines 86: which two sections?

-Section 2: change title in "Observation data" and you may want to move lines 73-85 here.

-Line 122: please specify which type of strainmeters you are working with.

-Line 175: how did you choose the window lenght? How does it influence your analysis?

-Line 183: can you explain or add references for the "expansion factor"? Also, is it the factor $d$ that you introduce on line 190?

-Line 201: please uniform the notation that you use for convolution with respect to equation 4.

-Section 3.2.3: do you mean "Spatio-temporal"?

-Line 254: what do you mean by "strain conversion equation"? Please be more specific. How have the strainmeters been calibrated? Did you carry out the calibration yourself or were you provided with the matrices?

-Figure 6: what are the units of y-axis?

-Line 260: how have you validated the results of the SVMD decomposition? What are the modes that you separated and how have you associated them with earthquake-related strain or other influencing factors?

-Line 265: have you looked only at surface strain or your methodology has been applied to shear strains as well? If not, please justify your choice.

-Figure 10b: can you comment on the further accelerations that we observe in this figure (e.g., around 2012.08.31)?

-Lines 328-330: maybe something like this to explain what are these anomalies could be anticipated in the text.

Line 363: have you made any inference about this anomaly-distance dependence? Do you think it is related to stress and/or fluids migration?

-Lines 365-375: following also my previous suggestion in General comment, have you compared your results with independent data? It is sometimes difficult to follow the causal relation between your observations and the inferences you make.

-Lines 379-386: since this analysis is preliminar to the definition of the anomalies it should be anticipated in the paper as well.

-Figure 12: I do not fully understand what this figures shows and what is the difference with respect to Figure 11.

**Typos:**

- Throughout the whole paper (e.g., lines 19, 24 and so on…) there is a dash among words where it should not be. Please double check the text.

-Line 46: pre-earthquake?

-Line 175: sliding step = shift ?

-Line 184: enabling to capture longer time series

-Line 341: Qiu et al. ?

-Line 361: situated farther or the farthest

---

## Author Comment (AC1)

**Response to Reviewer 1:**

I am very grateful to your comments for the manuscript. Thank you for your advice. All your suggestions are very important. They have important guiding significance for our paper and our research work. We have revised the manuscript according to your comments. The response to each revision is listed as following:

**Comment 1**

One drawback of this work is that it is applied to a single case study only. Why not applying to at least another case, in order to avoid that what is found is just associated to this unique case and cannot extend to other cases? If data are available, it would be interesting to compare with Wenchuan 2008 earthquake. This is done in Chi et al. 2023, but using just a single station.

By the way, regarding to this, there is another interesting paper on the comparison of the two case studies, although analysing different precursory parameters (from atmosphere): Liu et al. 2020, https://doi.org/10.3390/rs12101663.

**Response**:

Thanks for your suggestion.

In the process of our experiment, the data from Guza station began in 2006, the data from Xiaomiao and Luzhou station began in 2008, and the data from Zhaotong station began in 2010. Due to the lack of data on the Zhaotong station, in the case of using the same station and training data, the Wenchuan earthquake case is not suitable for comparative study with the Lushan earthquake case, so Wenchuan earthquake is not added to the submitted manuscript.

According to your suggestion, we used the same method to analyze the data before the Wenchuan earthquake and selected the data from the Guza, Luzhou, and Xiaomiao stations. Since the data began in 2008, we can only select 2010 and 2011 as the training set and validation set. The data from January to June 2008 were selected as the test set, and the method in the manuscript was used to analyze the pre-earthquake anomalies of the Wenchuan earthquake. The node diagram constructed by the distance between stations is shown in Fig. 1.

[Figure]

**Figure 1: Node diagram of the three borehole strain observation stations. The blue circle indicates the locations of the three borehole strain stations, the green line indicates the distance between the two stations, and the red star indicates the epicenter of the Wenchuan earthquake.**

We analyze the prediction results, use the definition in the manuscript to judge the abnormal days, and accumulate the abnormal days. Figure 2 shows the relationship between the accumulation of abnormal days and time at Guza, Luzhou, and Xiaomiao stations.

[Figure]

**Figure 2: Accumulation results of abnormal days of borehole data at Guza, Luzhou, and Xiaomiao stations. The dashed line indicates the date of the Wenchuan earthquake, while the red and blue curves indicate the results of the S-shaped fitting before and after the earthquake, respectively**

As shown in Fig. 2, the cumulative results of abnormal days at Guza station show the concavity of two parts. Some show that the abnormal accumulation accelerates from the beginning of January to April, the stress curve deviates from linearity, and the isolated area of strain release increases and extends steadily. The other part shows that the abnormal accumulation accelerated about a month before the earthquake, the strain release part on the fault accelerated expansion, and the strain level in the strain accumulation area increased rapidly. We fit the data of Xiaomiao station, and there is a similar phenomenon. It shows that the stations we selected receive more or less abnormal signals related to the Wenchuan earthquake. Our research is similar to the results of Chi et al., (2023) and Liu et al., (2020), which proves that the method in this paper is also applicable to the Wenchuan earthquake.

**Comment 2**

A second drawback is that it is not clearly explained the presence of the sigmoid in the results in terms of the physics of the earthquake preparation phase. Could you please interpret the results in terms of a physical model? Could it be related to a critical state of the regional crust? Could it be related to a dilatancy model of the lithosphere? How is the role of fluids?

**Response**:

  Thanks for your suggestion.

(1) "A second drawback is that it is not clearly explained the presence of the sigmoid in the results in terms of the physics of the earthquake preparation phase. Could you please interpret the results in terms of a physical model? Could it be related to a critical state of the regional crust? Could it be related to a dilatancy model of the lithosphere?"

The findings of this study align with the theory of the synergism process of a fault. Ma and Guo, (2014) conducted a laboratory modeling study on the instability of a planar strike-slip fault, suggesting that the occurrence of an earthquake is linked to a fault's synergistic process, which encompasses three stages. In the initial stage, there's a deviation of the stress curve from linearity. The second stage is marked by the steady increase and expansion of isolated areas of strain release. In the final stage, the fault's sections of strain release accelerate and expand, alongside a rapid increase in strain levels in areas of strain accumulation. The period from September to December 2012 corresponds to the first and the second stages, where the stress curve deviates from linearity and isolated areas of strain release grow and extend steadily. From early 2013 up to the earthquake, aligns with the third stage, characterized by the accelerated expansion of strain release sections on the fault and a swift rise in strain levels in strain-accumulation areas. The multitude of anomalies observed post-earthquake, including those caused by crustal fractures and aftershocks, were also evident. Similar phenomena were recorded at the XM and LZ stations, correlating with Ma's theory. Thus, we believe that the anomalous phenomena observed prior to the Lushan earthquake are related to the earthquake's gestation process.

[Figure]

**Figure 3: Accumulation results of anomalous days in borehole data from each station. The dashed line indicates the date of the earthquake, while the red and blue curves indicate the results of the S-shaped fitting before and after the earthquake, respectively. (a) Anomalous day accumulation results for Guza station; (b) Anomalous day accumulation results for Xiaomiao, Luzhou, and Zhaotong stations.**

(2) "How is the role of fluids?"

Borehole strain monitoring involves the placement of strain gauges deep underground to measure changes in rock or crustal strain. Crustal strain arises from the movement of tectonic plates and seismic activity. This method provides direct insights into the rate and pattern of crustal deformation, which is extremely helpful in understanding the stress state of the Earth's crust associated with seismic activities. Strain data are often highly sensitive to impending earthquakes, offering valuable information about potential fault planes.

Underground fluid monitoring primarily refers to tracking changes in groundwater levels, groundwater pressure, or the chemical composition of subterranean fluids. Seismic activities can affect the flow and pressure of groundwater, so monitoring these changes can indirectly detect seismic activities. Variations in underground fluids may

correlate with seismic activities, particularly preceding earthquakes. Anomalous fluctuations in groundwater levels and pressures can serve as precursors to earthquakes. Borehole strain data provide direct information on crustal strain, while underground fluid data offer indirect insights into fluid dynamics related to seismic activities. Both play crucial roles in earthquake precursor studies, yet they differ in their monitoring methodologies, sensitivities, and scopes of application.

**Comment 3**

Title. I suggest to add at the end of the title "(China)" since not all researchers know where Lushan is (especially who did not work on that earthquake).
**Response**:

Thanks for your suggestion.
Changed the title "Extraction of Pre-earthquake Anomalies in Borehole Strain Data Using Graph WaveNet: A Case Study of the Lushan Earthquake". Modify the title to "Extraction of Pre-earthquake Anomalies in Borehole Strain Data Using Graph WaveNet: A Case Study of the 2013 Lushan Earthquake, China".

**Comment 4**

Line 60. There are exceptions to the sentence "they mostly focused on single-station data": not only Liu et al. 2019 and Yu et al. 2020 (both already cited by Li et al.) but also Zhu et al. 2019 (Nonlinear Processes in Geophysics, https://doi.org/10.5194/npg-26-371-2019 not cited) to give a recent example of multi-station data analyses.
**Response**:

Thanks for your suggestion.
Modified "Despite the valuable insights gained from these studies, they mostly focused on single-station data, overlooking the potential correlations between multiple stations." and added " The study of seismic monitoring data based on multiple stations has been applied to many scenarios. Liu et al., (2019) analyzed the abnormal fluctuations of aerosol optical depth (AOD) before and after the 2008 Wenchuan earthquake and the 2013 Lushan earthquake, and found that the abnormal high AOD values appeared 11 days before the Wenchuan earthquake and 4 days before the Lushan earthquake. It is considered that the AOD index may be suitable as a precursor to the earthquake in the Sichuan Basin. Using borehole strain data from six stations in the Sichuan-Yunnan region, Yu et al., (2020) established a graph network and analyzed 13 earthquake cases with $Es > 10^7$ in the study area. It was found that the strain anomaly before the earthquake generally occurred within the first 30 days of the earthquake event. To study the abnormal strain changes before the Wenchuan earthquake, Zhu et al. (2019) introduced negative entropy analysis to the borehole data of three stations. The results show that Guza and Xiaomiao stations have similar trends and may record abnormal changes related to the Wenchuan earthquake. Renhe station failed to detect the anomalies before the earthquake due to the distance. An example of multi-station analysis is given, which shows that it is feasible to analyze seismic data with multi-station."

**Comment 5**

Figure 1 (and rest of the paper). The findings of the work are finally drawn in terms of accumulation of anomalies. This comprehensive way to express the results, in my knowledge, has been firstly proposed by De Santis et al. 2017 (https://doi.org/10.1016/j.epsl.2016.12.037) in a study of satellite magnetic field data in occasion of the large 2015 Nepal earthquake. In that paper, it was also introduced the notation "S-shape" for the first time, as it is also used in this paper (e.g. see Figure 10 caption).

**Response**:

  Thanks for your suggestion.

Modified " In Fig. 9, it is evident that the abnormal days we defined exhibit short-period, high-frequency oscillation signals in the original waveform, suggesting that these days are associated with crustal activity. " and added " Santis et al., (2017) study the 2015 Nepal event using Swarm magnetic satellite data. For the first time, an S-shaped fitting function was proposed in the abnormal accumulation analysis, and some abnormal differences were found in the area around the EQ epicenter from the abnormal accumulation results. By comparing the S-shaped function and the linear fitting, it was found that the S-shaped fitting was significantly better than the linear fitting. In this paper, the S-type function is used to fit the abnormal accumulation results. "

**Comment 6**

Line 175 and following. Why did you choose the window size of 7 days? How critical could this choice be?

**Response**:

  Thanks for your suggestion.

We choose the sliding window size standard from the equipment bearing capacity and the efficiency of data processing, through the experiment to select the optimal window size. As shown in Table 1 below, we selected the size of the sliding window for 7 days, 15 days, and 30 days, respectively. Table 1 gives the time and memory size required for the calculation process. If the size of the sliding window is too small, the correlation between the data cannot be maintained. Considering the time required for the SVMD calculation process and the memory size of the computer, we chose the size of the sliding window to be 7 days.

**Table 1.** The experimental results of SVMD correspond to different sliding window sizes.

| Window(day) | Time(min) | Memory(MB) |
|:---:|:---:|:---:|
| 7Days | 22.5 | 85.7 |
| 15Days | 51.6 | 171.1 |
| 30Days | 125.9 | 308.6 |

**Comment 7**

Line 242. Are you sure that std_error is the root mean square error? From the name it looks like the standard deviation error (the two quantities are different because of a slightly different denominator).

**Response**:

Thanks for your suggestion.

Removed std_error. In the process of the experiment, we use the root mean square error to calculate the upper and lower bounds of the predicted value. The std_error in line 242 and formula (11) have been modified to rmse.

**Comment 8**

There are section 5 (Results) and section 6 (Conclusion). What is missing is a section "Discussion", that is partly present in section 5.

**Response**:

Thanks for your suggestion.

We have modified the structure of the manuscript. In the fifth part, we mainly include the analysis of the prediction results, the analysis of the details of the randomly selected abnormal days, and the analysis of the abnormal accumulation results. The sixth part is added as the chapter of discussion, which mainly includes the comparison and discussion of the abnormal accumulation results between different stations and the elimination of the influence of meteorological factors. The seventh section contains the conclusion. And modify the 90 lines of the original manuscript "Section five mainly includes the analysis of prediction results, the detailed analysis of randomly selected abnormal days, and the analysis of abnormal accumulation results. The sixth part is the discussion, which mainly includes the comparison and discussion of the abnormal accumulation results between different stations and the exclusion of the influence of meteorological factors. The final section presents the conclusions of the study and summarizes the key insights drawn from our analysis." And modify the 391 lines of the original manuscript "Therefore, we can exclude the influence of pressure, temperature, and rainfall on the anomalies observed in the pre-earthquake borehole data from Lushan. We have reason to believe that the anomalies we extracted before the Lushan earthquake are related to the seismogenic process."

**Comment 9**

There are several words interrupted by a "-": e.g. "dam-age"(Line 24), "sur-face" (line 38), "phenome-non" (line 57), etc. Please join the two parts in just one.

**Response**:

Thanks for your suggestion.

Delete the "-". The "dam-age" in line 24 was modified to "damage", the "sur-face" in line 38 was modified to "surface", and the "phenome-non" in line 57 was modified to "phenomenon".

**Comment 10**

Line 86. "two sections": do you mean "next section"?

**Response**:

Thanks for your suggestion.

The "two sections" were deleted. The meaning you want to express here is the next section, and line 86 is changed to "next section".

**Comment 11**

Line 200 (equation (9)). Which is the "sigmod" function? Is it actually "sigmoid" as introduced in the line before?

**Response**:

Thanks for your suggestion.

Delete tanh and sigmod in equation (9). The tanh and sigmod in Equation (9) of line 200 are the activation functions of the neural network, tanh is the activation function of the output, and sigmod is the activation function that determines the information ratio transmitted to the next layer. The sigmod function in equation (9) is different from the sigmod function mentioned in the previous row. To avoid ambiguity in the symbol, the equation (9) in line 200 is modified to $T = g(W_1 * x + b_1) \bullet \sigma(W_2 * x + b_2)$, and in line 202 is added "where $g$ is the activation function of the output, $\sigma$ is the activation function that determines the ratio of information passed to the next layer."

**Comment 12**

Figure 9. The numbers at the axes are too small. Please enlarge them in order to let them more visible.

**Response**:

Thanks for your suggestion.

The value of the coordinate axis in Fig. 9 has been modified.

[Figure]

**Figure 9: Plots of raw data from four randomly selected anomalous days at each station.**

**References**

Chi, C., Li, C., Han, Y., Yu, Z., Li, X., and Zhang, D.: Pre-earthquake anomaly extraction from borehole strain data based on machine learning, Scientific Reports, 13, 10.1038/s41598-023-47387-z, 2023.

Liu, Q., De Santis, A., Piscini, A., Cianchini, G., Ventura, G., and Shen, X.: Multi-Parametric Climatological Analysis Reveals the Involvement of Fluids in the Preparation Phase of the 2008 Ms 8.0 Wenchuan and 2013 Ms 7.0 Lushan Earthquakes, Remote Sensing, 12, 10.3390/rs12101663, 2020.

Ma, J. and Guo, Y.: Accelerated synergism prior to fault instability: Evidence from laboratory experiments and an earthquake case, Dizhen Dizhi, 36, 547-561, 10.3969/j.issn.0253-4967.2014.03.001, 2014.

Liu, Q., Shen, X., Zhang, J., and Li, M.: Exploring the abnormal fluctuations of atmospheric aerosols before the 2008 Wenchuan and 2013 Lushan earthquakes, Advances in Space Research, 63, 3768-3776, 10.1016/j.asr.2019.01.032, 2019.

Yu, Z., Hattori, K., Zhu, K., Chi, C., Fan, M., and He, X.: Detecting Earthquake-Related Anomalies of a Borehole Strain Network Based on Multi-Channel Singular Spectrum Analysis, Entropy, 22, 10.3390/e22101086, 2020.

Zhu, K., Yu, Z., Chi, C., Fan, M., and Li, K.: Negentropy anomaly analysis of the borehole strain associated with the Ms 8.0 Wenchuan earthquake, Nonlin. Processes Geophys., 26, 371–380, 10.5194/npg-26-371-2019, 2019.

Santis, A. D., Balasis, G., Pavón-Carrasco, F. J., Cianchini, G., Mandea, M. J. E., and Letters, P. S.: Potential earthquake precursory pattern from space: The 2015 Nepal event as seen by magnetic Swarm satellites, 461, 119-126, 10.1016/j.epsl.2016.12.037, 2017.

---

## Author Comment (AC4)

**Response to Reviewer:**

I am very grateful to your comments for the manuscript. Thank you for your advice. All your suggestions are very important. They have important guiding significance for our paper and our research work. We have revised the manuscript according to your comments. The response to each revision is listed as following:

**General Comment**

The paper "*Extraction of Pre-earthquake Anomalies in Borehole Strain Data Using Graph WaveNet: A Case Study of the Lushan Earthquake*" presents a Graph Wavenet method to analyze the large amount of data collected at four borehole strainmeters before and after the Lushan Ms 7 earthquake. The authors highlight an acceleration of anomalies accumulation, and they infer a possible release of energy from a weak fault section and a strain accumultion on a strong section of the fault.

Although this work proposes a promising approach to analyze large chunks of data, two main drawbacks emerge in this reviewer opinion: (i) the paper does not flow optimally and it would need a better re-organization (see specific comments); (ii) it seems like the authors focused on the Graph Wavenet sometimes leaving behind a more accurate physical interpretation of the results obtained. As a suggestion, after having recognized the anomalous days, it would be interesting to have a more detailed discussion of these "anomalous data". Have you compared your results with different data (e.g., seismicity rates, pore-pressure data, deformations from GNSS measurements…)?

**Response**:

    Thanks for your suggestion.

(1) the paper does not flow optimally and it would need a better re-organization (see specific comments);

We have made changes to the manuscript process, which are in the responses to specific comments.

(2) it seems like the authors focused on the Graph Wavenet sometimes leaving behind a more accurate physical interpretation of the results obtained. As a suggestion, after having recognized the anomalous days, it would be interesting to have a more detailed discussion of these "anomalous data".

In response to specific comments, we have provided a more accurate physical interpretation of the results obtained and a more detailed discussion of these "anomalous data", and have added the interpretation and discussion to the original manuscript.

(3) Have you compared your results with different data (e.g., seismicity rates, pore-pressure data, deformations from GNSS measurements…)?

We compared our results with different data, and the comparisons and corresponding analysis are in the responses to specific comments.

**Specific Comments**
**Comment 1**

-Line 15: "acceleration of anomalous accumulation" of what? And in general I believe that the term "anomaly" should be better defined at the beginning of the paper.

**Response**:

Thanks for your suggestion.

In the original manuscript, we gave the definition of anomalies: for point anomalies in the prediction results of the borehole strain data, we gave the following judgment conditions: (a) there must be more than 15 points outside the interval within a 30-minute period; and (b) the difference between the centroid of the prediction interval and the actual value must be greater than 1.5 times the bandwidth of the interval, and there must be more than three such points in that 30-minute period. Days that satisfy the judgment conditions are defined as anomalous days.

De Santis et al., (2017) studied the 2015 Nepal event using Swarm magnetic satellite data, and for the first time proposed the S-shaped fitting function in the anomaly accumulation analysis. The most significant feature of the S-shaped fitting function is that it presents a curve in the shape of S. In this paper, we use the S-shaped function to fit the anomaly accumulation results. The "acceleration of anomalous accumulation" refers to the fact that the S-shaped curve of this anomaly accumulation grows slowly at the beginning, then grows rapidly in the middle, showing an accelerating effect, and eventually flattens out.

We have modified the abstract: we have modified "This study proposes using a graph wavenet graph neural network to analyze borehole strain data from multiple stations near the earthquake epicenter and establishes a node graph structure using data from four stations near the Lushan epicenter, covering years 2010–2013.", followed by the addition of "We define anomalies as follows: (a) detecting more than 15 points outside the intervals within a 30-minute window; (b) identifying difference between the center of predicted intervals and the actual values exceeding 1.5 times the bandwidth of the intervals, with more than three such points in the same 30-minute period. Days meeting these conditions were considered anomalous (Chi et al., 2023).".

**Comment 2**

-Line 48: which type of borehole strainmeters have been installed? Please specify…

**Response**:

Thanks for your suggestion.

Modified "China has deployed multiple four-component borehole strainmeters". It is modified to "China has deployed multiple YRY-4 four-component borehole strainmeters (Qiu et al., 2009)".

**Comment 3**

-Lines 73-85: maybe this is too premature here? There is no need to go into the details of the algorithm here. I'd re-organize the introduction moving the description of the earthquake (lines 95-100) before the state of art (before line 28). I'd also be more coincise on the description of previous works on this earthquake.

**Response**:

Thanks for your suggestion.

We have modified this explanation and adjusted it to a suitable position. More description of previous works on this earthquake has been modified to the manuscript.

**Comment 4**

-Lines 86: which two sections?

**Response**:

Thanks for your suggestion.

The "two sections" were deleted. The meaning we want to express here is the next section, and line 86 is changed to "next section".

**Comment 5**

-Section 2: change title in "Observation data" and you may want to move lines 73-85 here.

**Response**:

Thanks for your suggestion.

We have changed the title of Section 2 to " Observation data" and moved lines 73-85 to the appropriate place.

**Comment 6**

-Line 122: please specify which type of strainmeters you are working with.

**Response**:

Thanks for your suggestion.

Modified "This specialized strainmeter comprises four horizontally positioned sensors designed to measure changes in borehole diameter.". It is modified to "This specialized YRY-4 strainmeter comprises four horizontally positioned sensors designed to measure changes in borehole diameter (Qiu et al., 2009)".

**Comment 7**

-Line 175: how did you choose the window length? How does it influence your analysis?

**Response**:

Thanks for your suggestion.

We choose the sliding window size standard from the equipment bearing capacity and the efficiency of data processing, through the experiment to select the optimal window size. As shown in Table 1 below, we selected the size of the sliding window for 7 days, 15 days, and 30 days, respectively. Table 1 gives the time and memory size required for the calculation process. If the size of the sliding window is too small, the correlation between the data cannot be maintained. Considering the time required for the SVMD calculation process and the memory size of the computer, we chose the size of the sliding window to be 7 days.

**Table 1.** The experimental results of SVMD correspond to different sliding window sizes.

| Window(day) | Time(min) | Memory(MB) |
|-------------|-----------|------------|
| 7Days       | 22.5      | 85.7       |
| 15Days      | 51.6      | 171.1      |
| 30Days      | 125.9     | 308.6      |

**Comment 8**

-Line 183: can you explain or add references for the "expansion factor"? Also, is it the factor d that you introduce on line 190?

**Response**:

Thanks for your suggestion.

Expansion convolution, also known as cavity convolution or expansion convolution, is the addition of an expansion factor to the standard convolution kernel to increase the receptive field of the model. Unlike standard convolution, expansion convolution introduces a hyperparameter called "dilation rate", which is also known as "expansion factor", referring to the number of intervals between points in the convolution kernel (Yu and Koltun, 2015; Wang et al., 2018) The d we introduce in line 190 is the expansion factor.

**Comment 9**

-Line 201: please uniform the notation that you use for convolution with respect to equation 4.

**Response**:

Thanks for your suggestion.

Modified the symbol $*$ used for convolution in Eq. (4) to $\bullet$.

**Comment 10**

-Section 3.2.3: do you mean "Spatio-temporal"?

**Response**:

Thanks for your suggestion.

What we use here is "Spatio-temporal". Because in each "Spatio-temporal", both the temporal features of the borehole strain data and the spatial features between the borehole observation stations are extracted.

**Comment 11**

-Line 254: what do you mean by "strain conversion equation"? Please be more specific. How have the strainmeters been calibrated? Did you carry out the calibration yourself or were you provided with the matrices?

**Response**:

Thanks for your suggestion.

(1) How have the strainmeters been calibrated? Did you carry out the calibration yourself or were you provided with the matrices?

The new YRY-4 four-component borehole strainmeters independently developed by China have a data sampling rate of once per minute. The strainmeters adopt a double-ring lining model. Figure 1 gives a schematic diagram of the double-ring lining model for measuring borehole strain. The double-ring lining model assumes the linear elasticity and homogeneity of the medium and is used to measure the horizontal strain state of the rock. The diameter change of the corresponding azimuth angle $\theta_i$ caused by the change of the strain state is directly measured by the $i$ strainmeters in the cylinder (Chi et al., 2009). The relationship between the measured value $s_i$ and the

strain changes $(\varepsilon_1, \varepsilon_2, \varphi)$ is as follows :

$$s_i = A(\varepsilon_1 + \varepsilon_2) + B(\varepsilon_1 - \varepsilon_2)cos2(\theta_i - \varphi), \tag{1}$$

where $\varepsilon_1$ and $\varepsilon_2$ are the maximum and minimum principal strains, respectively, $\varphi$ is the principal direction, and $A$ and $B$ are the two parameters to be determined.

[Figure]

**Figure 1: Sketch of the two-ring system for measuring strain in boreholes (Qiu et al., 2013).**

The YRY-4 four-component borehole strainmeters contains four horizontally placed sensors, and the angle interval of the four sensors is 45°,

$$\begin{cases} s_1 = s_{\theta_1} = A(\varepsilon_1 + \varepsilon_2) + B(\varepsilon_1 - \varepsilon_2)cos2(\theta_1 - \varphi) \\ s_2 = s_{\theta_{1+\pi/4}} = A(\varepsilon_1 + \varepsilon_2) - B(\varepsilon_1 - \varepsilon_2)sin2(\theta_1 - \varphi) \\ s_3 = s_{\theta_{1+\pi/2}} = A(\varepsilon_1 + \varepsilon_2) - B(\varepsilon_1 - \varepsilon_2)cos2(\theta_1 - \varphi) \\ s_4 = s_{\theta_{1+3\pi/4}} = A(\varepsilon_1 + \varepsilon_2) + B(\varepsilon_1 - \varepsilon_2)sin2(\theta_1 - \varphi) \end{cases}, \tag{2}$$

$s_i (i = 1,2,3,4)$ are the measured values from the four sensors, and the YRY-4 four-component drilling strain gauge has good data self-testing(Su, 2019). The relationship between the measured values from the four sensors can be expressed as:

$$s_1 + s_3 = s_2 + s_4, \tag{3}$$

The Eq. (3) is a self-consistent equation for the YRY-4 four-component borehole strainmeters, which is considered reliable when the data satisfy the above results. Due to the different coupling between the probe and the surrounding rock and some other reasons, the four sets of measured values do not necessarily conform to the relationship of Eq. (3) completely, and it is necessary to correct the component measured values according to certain assumptions, Relative In Situ Calibration (Liu et al., 2011; Qiu et al., 2013). In this case, the

$$S_i = k_i s_i (i = 1,2,3,4), \tag{4}$$

Where $k_i (i = 1,2,3,4)$ is the calibration coefficient of each component and $S_i (i = 1,2,3,4)$ is the result after calibration of each component in turn. Assumption:

$$S_1 + S_3 = S_2 + S_4, \tag{5}$$

Bring Eq. (4) into Eq. (5):

$$k_1 s_1 + k_3 s_3 = k_2 s_2 + k_4 s_4, \tag{6}$$

The observed values of the four components are used to represent $s_i (i = 1,2,3,4)$. The calibration coefficient $k_i (i = 1,2,3,4)$ is obtained by inversion, and then the relative calibration results $S_i (i = 1,2,3,4)$ are obtained.

Because it is extremely complicated to calculate $A$ and $B$ through parameters such as the inner and outer diameters of the sleeve, the surrounding rock, the filled cement, and

the Young 's modulus and Poisson 's ratio of the sleeve material, the theoretical solid tide is used to test the observed solid tide, and $A$ and $B$ can be inverted. The inversion of coefficients $A$ and $B$ is called Absolute In Situ Calibration(Qiu et al., 2009; Qiu et al., 2013). If it is assumed that the theoretical strain solid tide is equal to the real strain solid tide, and the strain changes as $(\varepsilon_1, \varepsilon_2, \varphi)$, then the coefficients $A$ and $B$ can be obtained by the following two formulas :

$$\begin{cases} 2A(\varepsilon_1 + \varepsilon_2) = S_1 + S_3 = S_2 + S_4 = (S_1 + S_2 + S_3 + S_4)/2 \\ \quad\quad 4B^2(\varepsilon_1 - \varepsilon_2)^2 = (S_1 - S_3)^2 + (S_2 - S_4)^2 \end{cases}, \quad (7)$$

$A$ and $B$ are called absolute calibration coefficients.

The strainmeters are calibrated by the above relative in situ calibration and absolute in situ calibration.

(2) what do you mean by "strain conversion equation"? Please be more specific.

The plane strain state has only three independent components. The four-component borehole strainmeters have four components and record four sets of observations. The angle between the adjacent components of the probe of the four-component borehole strainmeters in China is 45 °, when the data satisfies the self-consistent equation, the observed data can be converted as follows:

$$\begin{cases} S_{13} = S_1 - S_3 = \varepsilon_{\pi/4} \\ S_{24} = S_2 - S_4 = -\varepsilon_0 \\ S_a = (S_1 + S_2 + S_3 + S_4)/2 \end{cases}, \quad (8)$$

The four-component borehole strainmeters observation data are converted into three indirect observations, and we call $(S_{13}, S_{24}, S_a)$ as alternative observations. Where $S_a$ corresponds to the surface strain, and $S_{13}$ and $S_{24}$ correspond to two independent shear strains. The "strain conversion equation" is to convert the values of the four components into one surface strain and two independent shear strains through the above calculation process.

**Comment 12**

-Figure 6: what are the units of y-axis?

**Response**:

Thanks for your suggestion.

The Y-axis of Fig. 6a is the value of the surface strain Sa, which is dimensionless; the Y-axis of Fig. 6b is the value of Earthquake-related Strain after SVMD preprocessed, which is also dimensionless.

**Comment 13**

-Line 260: how have you validated the results of the SVMD decomposition? What are the modes that you separated and how have you associated them with earthquake-related strain or other influencing factors?

**Response**:

Thanks for your suggestion.

In order to verify the decomposition results of our SVMD, we choose a month of surface strain data Sa to decompose through SVMD, and the decomposition results are shown in Fig. 2. Figure 2a is our original data, and Fig. 2b is the result of SVMD

decomposition.

[Figure]

Figure 2: (a) Sa raw data. (b) SVMD decomposition results of Sa.

[Figure]

Figure 3: The FFT periodogram of component IMF2.

As is shown in Fig. 2b, it can be clearly seen that Sa is decomposed into five components: IMF1, IMF2, IMF3, IMF4, and IMF5. Comparing Fig. 2a and IMF1, we find that IMF1 represents the trend term. We do the Fourier transform of the IMF2 component, the result is shown in Fig. 3, the frequency of the signal is mainly concentrated in $f_1=1.157\times10^{-5}$Hz and $f_2=2.232\times10^{-5}$Hz, these two frequencies correspond to the semi-diurnal wave and diurnal wave frequency of the earth tide respectively. Therefore, we believe that IMF2 corresponds to the influence of earth tides (Chi et al., 2019). We only remove the trend and solid tide, because the IMF3-IMF5 component contains a large number of strain signals, so we retain the remaining IMF3-IMF5 component as the research object.

**Comment 14**
-Line 265: have you looked only at surface strain or your methodology has been applied to shear strains as well? If not, please justify your choice.

**Response**:

Thanks for your suggestion.

In order to verify that our results are also applicable to the shear strain, we use the shear strain S13 data of Guza, Xiaomiao, Luzhou, and Zhaotong stations, and use the same method to analyze them. The abnormal day accumulation results of the shear strain S13 data of the four stations are shown in Fig. 4.

[Figure]

Figure 4: Accumulation results of anomalous days in S$_{13}$ borehole data from each station. The dashed line indicates the date of the earthquake, while the orange and purple curves indicate the results of the S-shaped fitting before and after the earthquake, respectively. (a) Anomalous day accumulation results for Guza station; (b) Anomalous day accumulation results for Xiaomiao, Luzhou, and Zhaotong stations.

[Figure]

Figure 5: Accumulation results of anomalous days in S$_a$ borehole data from each station. The dashed line indicates the date of the earthquake, while the red and blue curves indicate the results of the S-shaped fitting before and after the earthquake, respectively. (a) Anomalous day accumulation results for Guza station; (b) Anomalous day accumulation results for Xiaomiao, Luzhou, and Zhaotong stations.

We processed the data of shear strain S13 by using the same method, and the results are shown in Fig. 4. Figure 5 shows the processing results of surface strain Sa. By comparing Fig. 4 with Fig. 5, we find that the processing results of shear strain S13 and surface strain Sa are very similar. According to the above Eq. (8), we can see that the surface strain Sa can better reflect the four component data measured by the YRY-4 borehole strainmeters. Therefore, the data characteristics of the surface strain Sa are used as the research object in this paper.

**Comment 15**

- Figure 10b: can you comment on the further accelerations that we observe in this figure (e.g., around 2012.08.31)?

**Response**:

Thanks for your suggestion.

[Figure]

**Figure 6: Cumulative results of anomalous days in the time around 2012.08.31 at Xiaomiao, Luzhou and Zhaotong stations. The dotted line represents the date of the earthquake, and the purple curve represents the S-shaped fit.**

We did S-shape fitting for the anomalies in the time period around 2012.08.31, and the fitting results are shown in Fig. 6. As can be seen from the fitting result graph, the S-shaped curves of the three stations are almost all straight lines, and there is no obvious acceleration phenomenon. Therefore, we believe that the acceleration phenomenon in the time period around 2012.08.31 is not an anomaly related to the Lushan earthquake.

**Comment 16**

-Lines 328-330: maybe something like this to explain what are these anomalies could be anticipated in the text.

**Response**:

Thanks for your suggestion.

In the original manuscript, in the section on results, we first analyzed the prediction results obtained by the method we used, then constructed prediction intervals based on the prediction results, then gave the definition of anomalies, and based on the prediction intervals and the definition of anomalies, we determined which days are anomalies, and finally, we did an S-shape fitting of anomalous days and analyzed the results of the fitting.

Figure 9 in our original manuscript is not used to explain which specific type of anomaly pattern we extracted, but rather to verify that similar short-period, high-frequency oscillating seismic wave signals do occur on all of the anomalous days we extracted.

**Comment 17**

Line 363: have you made any inference about this anomaly-distance dependence? Do you think it is related to stress and/or fluids migration?

**Response**:

Thanks for your suggestion.

(1) have you made any inference about this anomaly-distance dependence?

Relevant research scholars have analyzed the detection capability of borehole strainmeters and given empirical equations for the relationship between the magnitude of the earthquake and the anomaly time of the short-term precursor of strain, as shown in Eq. (9) below; and the empirical equation between the magnitude of the earthquake and the monitoring range of the short-term precursor of the borehole strain gauges, as shown in Eq. (10) below (Su, 1991).

$$logT = 0.60M - 0.92 \tag{9}$$

$$logR^{(0)} = 0.182M + 1.4 \tag{10}$$

Where $T$ denotes the anomaly time of the strain short-term precursor, $M$ denotes the magnitude, and $R$ denotes the monitoring range of the short-term precursor of the borehole strainmeters. According to the above empirical Eq. (9) and Eq. (10), we substitute the magnitude Ms7.0 for this study, and obtain that the anomaly time of strain short-term precursors for an earthquake of magnitude 7.0 is about 159 days, and the monitoring range of the short-term precursors of the drilling strainmeters is about 469 km. The distances of the four stations selected for this study from the epicenter of the earthquake are 73 km, 268 km, 286 km, and 337 km, which indicate that all the stations we selected are capable of receiving the short-term precursors of the drilling strainmeters. By analyzing our experimental results, we find that the anomalies extracted in this paper were about 6 months before the earthquake, which is about the same as the time calculated by the above empirical formula, indicating that we also extracted the short-term precursor anomalies of strain.

For our extraction of this anomaly-distance dependence, there is no inference in this regard, and we cannot give specific conclusions about this anomaly-distance dependence. We can only combine the above time-magnitude and distance-magnitude empirical equations to analyze. In the follow-up work, we will try to quantify this anomaly-distance dependence.

(2) Do you think it is related to stress and/or fluids migration?

The anomalous patterns shown in the borehole stress-strain pre-seismic coincide with the results of the petrological experiments. Rock rupture experiments and theoretical studies show that when the stress accumulation in the pregnant seismic zone enters the nonlinear accumulation stage from linear accumulation, with further enhancement of the degree of stress and strain accumulation, the resulting stress perturbation will lead to changes in the additional stress and strain states at nearby strain measurement points. When the rock rupture strength reaches the critical destabilization stage, pre-slip occurs before the fault stick-slip, and the resulting stresses will lead to obvious anomalous responses at nearby stations. After entering the destabilization stage, i.e., the critical seismic time, the stress pattern before rupture of the adjacent rocks shows obvious tension and compression regions, which are related to local extension and weakening,

and most of the anomalies appear in the form of sudden jumps (Zhao et al., 1997; Ma et al., 1998; Li, 2002).

The influence of subsurface fluids on faults is mainly manifested in two aspects: first, the hydrodynamic effect of the upper crust, that is, fluid permeability and fluidity; and second, the effect of magma flow bubble diffusion and magma intrusion in the lower crust. The dynamic changes of underground fluid flow and permeability will affect the pore pressure of the faults, thus affecting the frictional instability of the faults (Zhang et al., 2007). Whether in the tensional subsidence zone or extrusion uplift zone, the rapid misalignment of the rock mass is accompanied by the rapid transportation of fluids. Fluid-bearing layers are energy storage layers, and the presence and transportation of fluids in faults reduce the friction between rock masses (Xu, 2002). Borehole strain observations allow direct observation of stress and strain in subsurface rocks, and the variability between rock bodies is related to the dynamics of subsurface fluids.

In summary, we believe that the extraction of this anomaly-distance dependence is related to both stress and/or fluids migration.

**Comment 18**

-Lines 365-375: following also my previous suggestion in General comment, have you compared your results with independent data? It is sometimes difficult to follow the causal relation between your observations and the inferences you make.

**Response**:

Thanks for your suggestion.

Due to the problem of data permissions, we could not obtain the corresponding seismicity rates, pore-pressure data, GNSS, and other data. We compared the findings of other data with ours by consulting the related literatures.

(1) following also my previous suggestion in General comment, have you compared your results with independent data?

Using the seismic activity data observed by the Qiaojia array from March 2012 to July 2014, Li et al., (2017) analyzed the anomalous characteristics of seismic activity in the observation area of the Qiaojia array before the Lushan earthquake. On the one hand, the lower limit of magnitude M=2.0 is selected. The analysis results show that the increase process of earthquake frequency before the Lushan earthquake occurred from September to November 2012 and lasted for about 3 months. On the other hand, taking the lower limit of magnitude M=0.3, with the focal depth h=10km as the boundary, the earthquake is divided into two parts. The results show that the frequency of earthquakes with focal depth h≤10km and h>10km before the Lushan earthquake increases obviously with time. From September 2012 to January 2013, the duration is about five and a half months. After January 20, 2013, the frequency of earthquakes with h≤10km is still increasing with time, while the frequency of earthquakes with h>10km decreases with time. According to the research results, they believe that it is because the Qiaojia array is located on the eastern boundary of the Sichuan-Yunnan rhombic block, and the Sichuan-Yunnan rhombic block is part of the eastern part of the Qinghai-Tibet Plateau. The observed increase in seismic activity is actually due to the enhancement of the

eastward movement of the eastern part of the Qinghai-Tibet Plateau. It is the eastward movement of the eastern part of the Qinghai-Tibet Plateau that led to the occurrence of strong earthquakes in the Sichuan-Yunnan region.

Xu et al., (2019) used the GNSS observation network to study the deformation before the Lushan earthquake. It was found that the deformation characteristics of the time series of a single station were not obvious, but the cumulative displacement of the entire GNSS network showed obvious abnormalities in the locking state and negative acceleration. The formation of the locking zone is mainly due to the increase of friction in the southern section of the Longmenshan fault zone and the decrease in the ability to resist deformation, resulting in smaller and smaller displacement until zero. The locking state lasted from 5 months before the earthquake to 2 months after the earthquake, and the locking state lasted for 7 months. The results of laboratory rock fracture experiments further confirmed this. However, in the locked state, the strain energy is still accumulating. Once the strain accumulation exceeds the frictional strength of the fault, the fault will rupture and release the accumulated strain energy. They described the entire evolution process as : stable linear rise process, locked state, earthquake occurrence, post-earthquake adjustment for about two months, and recovery of linear rise. They believe that the Lushan earthquake was due to the strong sine shear tectonic force to accelerated the earthquake preparation.

Conclusion: Li et al., (2017) analyzed the pre-seismic anomalies of the Lushan earthquake using seismic rate data. The synthesized results show that there is an anomaly of increasing earthquake frequency over time that lasts for 3-5 months from September 2010, which is highly consistent with our extracted anomaly of rapidly increasing anomalous days from October 2012 to January 2013, indicating that short-term precursor anomalies related to the Lushan earthquake were extracted by both seismicity rates and borehole strain data. Xu et al., (2019) studied the deformation before the Lushan earthquake using data from the GNSS observation network and found that a locking state, which started 5 months before the earthquake and lasted until 2 months after the earthquake, occurred before the Lushan earthquake, and that the strain energy was still accumulating under the locking state until the fault ruptured. Our results show that the anomaly duration lasted from six months before the earthquake to two months after the earthquake. By analyzing the GNSS observation network data and the borehole strain data, we concluded that because the strain energy was still accumulating, a partial pre-release of energy still occurred even in the locked state.

(2) It is sometimes difficult to follow the causal relation between your observations and the inferences you make.

We reviewed more research on the Lushan earthquake and used other observations to support the accuracy of our observations. We also give a more detailed description of the causal relationship between our observations and inferences. We put the corresponding description in the Results section of the manuscript.

**Comment 19**
-Lines 379-386: since this analysis is preliminar to the definition of the anomalies it should be anticipated in the paper as well.

**Response**:

Thanks for your suggestion.

In the second section, we added: "Despite its advantages, such as high sensitivity and a wide frequency band during observation, the four-component borehole strainmeter remains susceptible to interference from surrounding sources. We used the improved VMD algorithm to analyze the Sa data and found that the first two components in the decomposition results correspond to the annual trend term and the solid tide, respectively, and the remaining components contain a large number of strain signals. We retained the remaining components as research data. Because there is no ability to extract meteorological factors such as air pressure, temperature, and rainfall from the remaining components, we analyze the measured data of meteorological factors to determine whether the meteorological data affects the results of borehole strain observation."

**Comment 20**

-Figure 12: I do not fully understand what this figures shows and what is the difference with respect to Figure 11.

**Response**:

Thanks for your suggestion.

Figure 11 shows that there are obvious annual trends in air pressure and temperature, and rainfall is concentrated in summer. Figure 12 shows the differential processing of the above pressure, temperature, and rainfall data. The periodic changes can be removed by differential processing, which highlights the anomaly of the data. There are two time periods for the anomalies extracted from the Lushan earthquake: the first period is from October 2012 to January 2013; and the second period is from March to June 2013. It can be seen from the two images that no matter whether differential processing is performed, there are no abnormal changes in air pressure, temperature, and rainfall data for the time period when we extract anomalies, indicating that the abnormal signals we extract have nothing to do with these influencing factors.

**Typos:**

**Comment 1**

- Throughout the whole paper (e.g., lines 19, 24 and so on…) there is a dash among words where it should not be. Please double check the text.

**Response**:

Thanks for your suggestion.

Delete the "-". The "con-ducting" in line 19 was modified to "conducting", the "dam-age" in line 24 was modified to "damage", the "sur-face" in line 38 was modified to "surface", the "phenome-non" in line 57 was modified to "phenomenon", the "vali-dation" in line 82 was modified to "validation", the "geo-logical" in line 100 was modified to "geological", the "band-width" in line 145 was modified to "bandwidth", the "ap-plied" in line 246 was modified to "applied", the "earth-quake" in line 324 was modified to "earthquake", the "pre-ceding" in line 343 was modified to "preceding",

the "record-ed" in line 360 was modified to "recorded", the "rain-fall" in line 383 was modified to "rainfall", the "dis-playing" in line 383 was modified to "displaying", the "da-ta" in line 416 was modified to "data", and the "Pro-gram" in line 419 was modified to "Program".

**Comment 2**
-Line 46: pre-earthquake?
**Response**:
  Thanks for your suggestion.
Deleted "pro-earthquake". Changed to "pre-earthquake".

**Comment 3**
-Line 175: sliding step = shift ?
**Response**:
  Thanks for your suggestion.
The "sliding step" in line 175 refers to the step we choose to move the sliding window, not the "shift".

**Comment 4**
-Line 184: enabling to capture longer time series
**Response**:
  Thanks for your suggestion.
Deleted "enabling the capture of longer time-series". Changed to "enabling to capture longer time series".

**Comment 5**
-Line 341: Qiu et al. ?
**Response**:
  Thanks for your suggestion.
Deleted "Qiu et al. (An et al., 2013; Jiang et al., 2013; Yu et al., 2020; Qiu et al., 2013; Zhu et al., 2018) conducted research on short-term impending earthquake anomalies pre-ceding the Lushan earthquake and suggested that these anomalies occurred within a few days to one month before the earthquake.". Changed to "Relevant researchers have studied the short-term anomalies before the upcoming Lushan earthquake and believe that these anomalies occurred within a few days to a month before the earthquake(An et al., 2013; Jiang et al., 2013; Yu et al., 2020; Qiu et al., 2013; Zhu et al., 2018).".

**Comment 6**
- Line 361: situated farther or the farthest
**Response**:
  Thanks for your suggestion.
Deleted "situated farthest from the epicenter,". Changed to "situated farther from the epicenter,".

**References**

Chi, C., Li, C., Han, Y., Yu, Z., Li, X., and Zhang, D.: Pre-earthquake anomaly extraction from borehole strain data based on machine learning, Scientific Reports, 13, 10.1038/s41598-023-47387-z, 2023.

Chi, C., Zhu, K., Yu, Z., Fan, M., Li, K., and Sun, H.: Detecting Earthquake-Related Borehole Strain Data Anomalies With Variational Mode Decomposition and Principal Component Analysis: A Case Study of the Wenchuan Earthquake, IEEE Access, 7, 157997-158006, 10.1109/access.2019.2950011, 2019.

Chi, S., Chi, Y., Deng, T., Liao, C., Tang, X., and Chi, L.: The Necessity of Buiding National Strain-Observation Network from the Strain Abnormality Before Wenchuan Earthquake, Recent Developments in Word Seismology, 1-13, 2009.

De Santis, A., Balasis, G., Pavón-Carrasco, F. J., Cianchini, G., and Mandea, M.: Potential earthquake precursory pattern from space: The 2015 Nepal event as seen by magnetic Swarm satellites, Earth and Planetary Science Letters, 461, 119-126, 10.1016/j.epsl.2016.12.037, 2017.

Li, S.: A Study of Mid-Term and Short-Impending Earthquake Prediction Method of Borehole Strain-Stress Anomalies, Bulletin of the Institute of Crustal Dynamics, 115-126, 2002.

LI, Y., Chen, L., and Chen, X.: Enhancement of Seismicity Recorded by the Qiaojia Seismic Network before the 2013 Lushan and 2014 Ludian Earthquakes, EARTHQUAKE, 37, 95-106, 2017.

Liu, Q., Zhang, J., and Chi, S.: Quality Evaluation and Fitting Analysis of 4-component Borehole Strainmeter Data, EARTHQUAKE, 31, 87-96, 2011.

Ma, J. and Guo, Y.: Accelerated synergism prior to fault instability: Evidence from laboratory experiments and an earthquake case, Dizhen Dizhi, 36, 547-561, 10.3969/j.issn.0253-4967.2014.03.001, 2014.

Ma, S., Liu, L., Ma, J., Liu, T., and Wu, X.: Experimental study of physical processes of rock fracture and friction and of destabilizing nucleation, Seventh Academic Conference of the Seismological Society of China, Jinggangshan, Jiangxi, China, 1.

Qiu, Z., Kan, B., and Tang, L.: Conversion and application of 4-component borehole strainmeter data, EARTHQUAKE, 29, 83-89, 2009.

Qiu, Z., Tang, L., Zhang, B., and Guo, Y.: In situ calibration of and algorithm for strain monitoring using four-gauge borehole strainmeters (FGBS), Journal of Geophysical Research: Solid Earth, 118, 1609-1618, 10.1002/jgrb.50112, 2013.

Su, K.: EARTHQUAKE-MONITORING CAPABILITY OF BOREHOLE STRAINMETER, EARTHQUAKE, 38-46, 1991.

Su, K.: Analysis of Surface Strain and Shear Strain from Four Component Borehole Strain Observation Data, EARTHQUAKE RESEARCH IN SHANXI, 30-35+52, 2019.

Wang, P., Chen, P., Yuan, Y., Liu, D., Huang, Z., Hou, X., and Cottrell, G.: Understanding convolution for semantic segmentation, 2018 IEEE winter conference on applications of computer vision (WACV), 1451-1460, 10.48550/arXiv.1702.08502.

Xu, C.: The effect of deep seated fluids on earthquake preparation and occurence, SOUTH CHINA JOURNAL OF SEISMOLOGY, 1-10, 10.13512/j.hndz.2002.03.001, 2002.

Xu, K., Gan, W., and Wu, J.: Pre-seismic deformation detected from regional GNSS observation network: A case study of the 2013 Lushan, eastern Tibetan Plateau (China), Ms 7.0 earthquake, Journal of Asian Earth Sciences, 180, 10.1016/j.jseaes.2019.05.004, 2019.

Yu, F. and Koltun, V.: Multi-scale context aggregation by dilated convolutions, ICLR 2016, 10.48550/arXiv.1511.07122.

Zhang, S., Yang, X., and Lu, Y.: Pore-Pressure Diffusion and Seismic Stress Triggering, 2007 Seismic Fluids Symposium, Yichang, Hubei, China, 1.

Zhang, X., Liu, X., Qin, S., and Jia, P.: Precursory Characteristics of Meta-Instability of Cross-Fault Deformation Before the Lushan Ms 7.0 Earthquake, Geomatics and Information Science of Wuhan University, 45, 1669-1677, 10.13203/j.whugis20190467, 2020.

Zhao, S., Wang, M., and Wei, H.: STUDY ON THE ANOMALY CHARACTERISTICS OF STRAIN AND STRESS MEASUREMENT BEFORE EARTHQUAKE, EARTHOUAKE RESEARCH IN PLATEAU, 51-57, 1997.

---

## Author Response (AR2)

**Response to Reviewer:**

I am very grateful to your comments for the manuscript. Thank you for your advice. All your suggestions are very important. They have important guiding significance for our paper and our research work. We have revised the manuscript according to your comments. The response to each revision is listed as following:

**Suggestions for revision**

The revision is fine, with the following exception: Line 345 and 553. Please change "Santis" with "De Santis". Obviously in the references list this citation should be moved in the correct alphabetic order.

**Response**:

Thanks for your suggestion.

We have changed "Santis" to "De Santis" in lines 345 and 553. The reference in line 553 has also been moved to the appropriate position in alphabetic order.